# Exploiting functional regions in the viral RNA genome as druggable entities

**Dehua Luo**[1,2,3,4†], **Yingge Zheng**[1,2,3,5†], **Zhiyuan Huang**[1,2,6†], **Zi Wen**[1,2,6], **Lijun Guo**[1,2,3], **Yingxiang Deng**[1,2,3], **Qingling Li**[1,2,3], **Yuqing Bai**[1,4], **Shozeb Haider**[7,8,9], **Dengguo Wei**[1,2,3,4,10,11]*

[1]National Key Laboratory of Agricultural Microbiology, College of Veterinary Medicine, Huazhong Agricultural University, Wuhan, China; [2]Hubei Hongshan Laboratory, Interdisciplinary Sciences Institute, Huazhong Agricultural University, Wuhan, China; [3]National Reference Laboratory of Veterinary Drug Residues (HZAU) and National Safety Laboratory of Veterinary Drug (HZAU), MOA Key Laboratory for Detection of Veterinary Drug Residues, MOA Laboratory for Risk Assessment of Quality and Safety of Livestock and Poultry Products, Wuhan, China; [4]College of Life Science and Technology, Huazhong Agricultural University, Wuhan, China; [5]Frontiers Science Center for Animal Breeding and Sustainable Production, Wuhan, China; [6]College of Informatics, Huazhong Agricultural University, Wuhan, China; [7]University of Tabuk (PFSCBR), Tabuk, Saudi Arabia; [8]UCL School of Pharmacy, University College London, London, United Kingdom; [9]UCL Centre for Advanced Research Computing, University College London, London, United Kingdom; [10]Shenzhen Institute of Nutrition and Health, Huazhong Agricultural University, Wuhan, China; [11]Shenzhen Branch, Guangdong Laboratory for Lingnan Modern Agriculture, Genome Analysis Laboratory of the Ministry of Agriculture, Agricultural Genomics Institute at Shenzhen, Chinese Academy of Agricultural Sciences, Shenzhen, China

**\*For correspondence:**
dgwei@mail.hzau.edu.cn

†These authors contributed equally to this work

## eLife Assessment

The study presents a **fundamental** advance in antiviral RNA research by adapting SHAPE-Map to chart the secondary structure of the porcine epidemic diarrhea virus (PEDV) genome in infected cells and pinpointing structurally conserved, accessible RNA elements as therapeutic targets. A broad, well-documented integration of biochemical probing, computational analysis, and functional validation provides **convincing** evidence that these regions are both biologically relevant and druggable. Beyond PEDV, the work offers a generalizable framework for RNA-guided antiviral discovery that will interest researchers in RNA therapeutics and viral genome biology.

**Abstract** RNA-targeting compounds or small interfering RNAs (siRNAs) offer a potent means to control viral infections. An essential prerequisite for their design depends on identifying conserved and functional viral RNA structures in cells. Techniques that probe RNA structures in situ have recently been developed including SHAPE-MaP, which has been helpful in the analysis of secondary structures of RNA. In this study, we report on the application of SHAPE-MaP to the porcine epidemic diarrhea virus RNA genome to categorize different functional regions, including potential quadruplex-forming sequence and target sites of siRNA. Our results show that these structures can be exploited to inhibit viral proliferation and that SHAPE-MaP is an effective method to identify secondary structures in RNA genomes.

## Introduction

RNA-targeting drugs are expected to address the challenges posed by 'undruggable' protein targets and significantly expand the landscape of targetable macromolecules (*Disney, 2019*, *Kovachka et al., 2024*, *Li and Wang, 2019*, *Ursu et al., 2020*, *Warner et al., 2018*). The US Food and Drug Administration (FDA) has approved six siRNAs (*Tang and Khvorova, 2024*), nine antisense oligonucleotides (ASOs) (*Bennett, 2019*, *Chen et al., 2024*, *Crooke et al., 2021*), and the RNA-targeting compound Risdiplam (*Ratni et al., 2021*, *Sheridan, 2021*) for treating complicated diseases such as spinal muscular atrophies. The selection of RNAs that are functional, conserved, and accessible to be targeted in vivo is fundamental to the development of RNA-targeting drugs. Unlike RNAs in vitro, the folding of RNAs in cells is affected by the intricate intracellular environment (*Manfredonia et al., 2020*, *Zhang et al., 2021b*). Thus, in situ probing for the accessibility of viral RNAs by antivirals is essential for the identification of targetable RNAs.

In recent years, the integration of chemical probing methods and high-throughput sequencing technologies has been used to analyze the structure of the viral RNA genome in infected cells (*Boerneke et al., 2019*, *Cao et al., 2021*, *Flynn et al., 2016*, *Huber et al., 2019*, *Huston et al., 2021*, *Lan et al., 2022*, *Li et al., 2018*, *Manfredonia et al., 2020*, *Mauger et al., 2015*, *Pollom et al., 2013*, *Siegfried et al., 2014*, *Smola et al., 2016*, *Yang et al., 2021*, *Zhang et al., 2021a*, *Ziv et al., 2020*). These methods have confirmed the widespread presence of structurally organized RNA elements within the viral genome, which are composed of various types of RNA structures such as stem loop structures (with double-helical stems), RNA single-strands, RNA pseudoknots, and RNA G-quadruplexes (G4s), collectively forming a complex and dynamic global genome architecture (*Boerneke et al., 2019*, *Kwok et al., 2016*). Of these, RNA G4 has shown considerable promise due to its high stability and ability for modulation by small molecules (*Zhao et al., 2021*, *Fang et al., 2023*, *Fleming et al., 2016*, *Kwok et al., 2016*, *Ruggiero and Richter, 2018*, *Wang et al., 2016*, *Zhang et al., 2021a*). Furthermore, RNA interference (RNAi) has been shown to inhibit gene expression and viral replication (*Bennett, 2019*, *Bowden-Reid et al., 2023*, *Piasecka et al., 2020*, *Qureshi et al., 2018*, *Sagan et al., 2010*, *Tang and Khvorova, 2024*, *Westerhout and Berkhout, 2007*). Despite advances, it remains a challenge to identify RNA elements or regions that serve as targets within complex genomic structural contexts.

Selective 2'-hydroxyl acylation and primer extension mutational profile (SHAPE-MaP) stands out in ex vivo RNA structure probe technologies for its ability to quantify nucleotide flexibility and solvent accessibility at single nucleotide resolution (*Lorenz et al., 2016*, *Siegfried et al., 2014*, *Smola et al., 2016*, *Lucks et al., 2011*). In short, SHAPE reagents like NAI selectively modify flexible unpaired 2'-OH groups in RNA, and these modifications are detected as mutations during reverse transcription, allowing precise mapping of RNA secondary structures by sequencing (*Siegfried et al., 2014*). In the viral long-stranded RNA genome, SHAPE reactivities are used as constraints in the prediction of RNA structures based on nearest-neighbor rules (*Deigan et al., 2009*, *Mathews, 2004a*), which can result in thousands of possible conformations. The pairing probability of each nucleotide derived from the SHAPE reactivities was subsequently used to calculate the Shannon entropy. Regions with high Shannon entropy can adopt alternative conformations, while those with low Shannon entropy correspond to well-defined RNA structures or persistently single-stranded regions (*Mathews et al., 2004b*, *Siegfried et al., 2014*). Therefore, SHAPE reactivity provides information about the pairing probability and accessibility of nucleotides, whereas Shannon entropy reflects the likelihood of alternative conformations in a given region. RNA motifs with low SHAPE reactivity and low Shannon entropy are assumed to be well-folded structures and potential targets (*Boerneke et al., 2019*). In this study, we categorized RNAs with distinct features in the viral genome according to SHAPE reactivity and Shannon entropy and investigate their potential as targets (*Figure 1*).

The porcine epidemic diarrhea virus (PEDV) is an alpha-coronavirus with a positive single-stranded RNA genome of 28 kb (*Jung et al., 2020*, *Lee, 2015*). It results in severe diarrhea and vomiting, leading to the death of 90% of infected piglets. The spread of PEDV is worldwide and its infection is of primary concern in pig husbandry (*Lee, 2015*). The epizootic reemergence of PEDV has also been reported to infect human intestinal cells and shows a potential susceptibility to infection by different species (*Niu et al., 2023*).

In this study, we adapted the SHAPE-MaP method to detect the in situ structure of the PEDV RNA genome in infected cells. More specifically, the combination of SHAPE reactivity and Shannon entropy

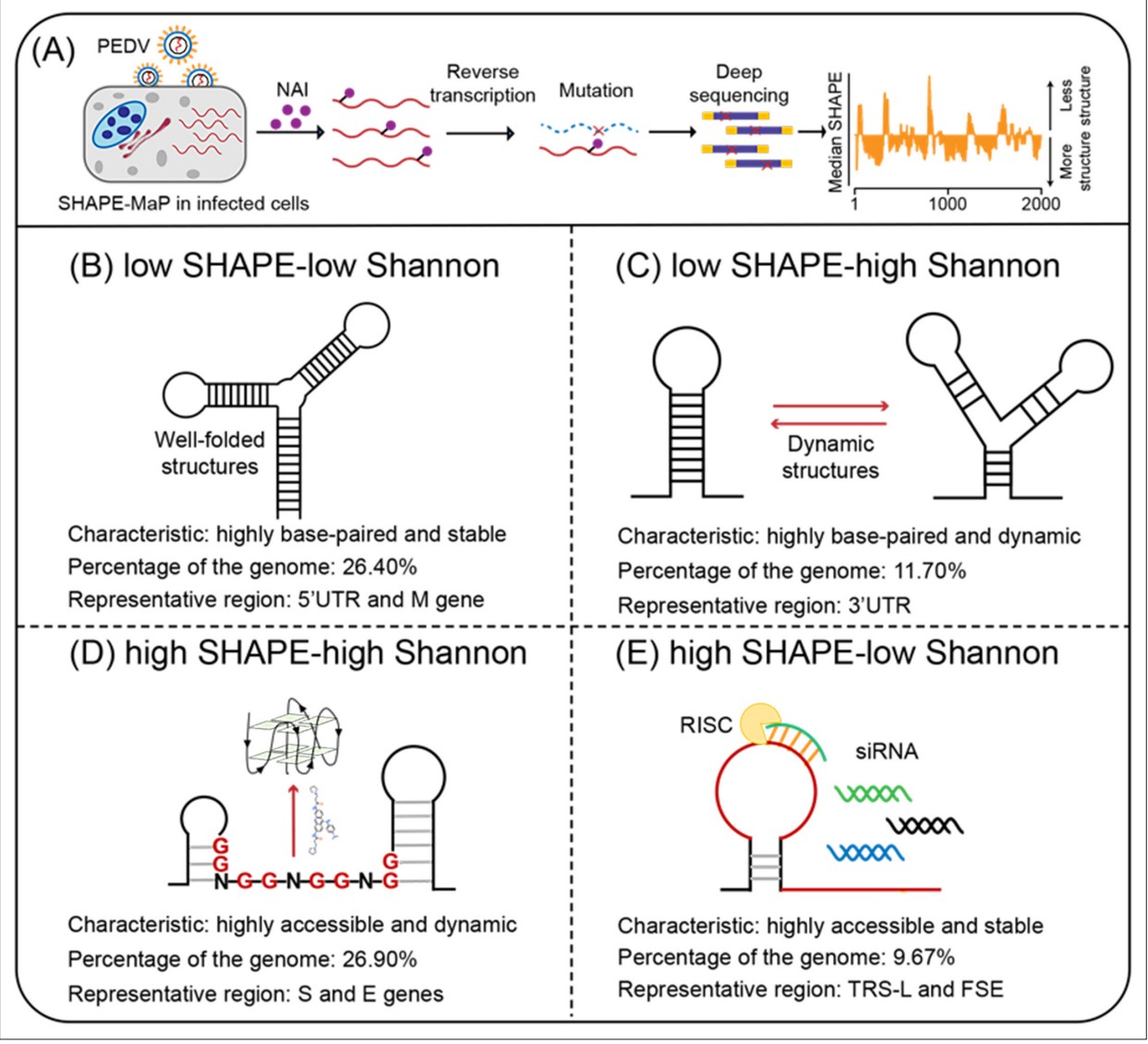

**Figure 1.** Four regions with different characteristics in the porcine epidemic diarrhea virus (PEDV) genome. (**A**) Schematic of SHAPE-MaP for probing the RNA structure of the PEDV genome in situ. (**B**) Well-folded regions (low SHAPE reactivity and low Shannon entropy; 26.40% of genome). These regions represent stably folded RNA structures with minimal conformational flexibility, serving as structural scaffolds or functional elements in viral replication. (**C**) Dynamic structured regions (low SHAPE reactivity and high Shannon entropy; 11.70% of genome). These conformationally plastic domains mediate regulatory switches between alternative secondary structures during infection. (**D**) Dynamic unpaired regions (high SHAPE reactivity and high Shannon entropy; 26.90% of genome). These regions are prone to form non-canonical nucleic acid structures (e.g., G-quadruplexes), which can be stabilized by small-molecule ligands to inhibit viral replication. (**E**) Persistent unpaired regions (high SHAPE reactivity and low Shannon entropy; 9.67% of genome). These regions are more accessible for siRNA binding, facilitating recruitment of Argonaute proteins and Dicer to form the RNA-induced silencing complex (RISC) for targeted cleavage.

The online version of this article includes the following figure supplement(s) for figure 1:

**Figure supplement 1.** Genome-wide SHAPE-MaP analysis of porcine epidemic diarrhea virus (PEDV).

categorized the PEDV RNA genome into different functional types (*Figure 1*), based on distinctive profiles, including the 5' untranslated region (5' UTR), frameshifting stimulatory element (FSE), and 3' untranslated region (3' UTR). We particularly focused on guanine-rich regions, which could potentially fold into a stable G4 structure and sites suitable for binding small interfering RNA (siRNA). Our results show that G4 folding in a dynamic single-stranded region with high SHAPE reactivity and high Shannon entropy inhibited viral proliferation, while siRNA targeting persistent single-stranded regions with high SHAPE reactivity and low Shannon entropy exhibited greater success rates in inhibiting viral

replication. Finally, this work proposes a strategy for selecting targetable RNAs based on the in situ folding characteristics of the viral genome.

## Results

### Mapping the RNA structures in the PEDV genome

SHAPE-MaP was used to characterize the structural features of the PEDV RNA genome in infected cells (*Figure 1A*). The high correlations of mutational signatures for biological replicates (ex vivo SHAPE R2=0.955; *Figure 1—figure supplement 1A*) and the high mutation rate with NAI treatment (*Figure 1—figure supplement 1B*) indicate the high quality of the sequencing data and the experimental workflow. Hence, two replicates were combined for downstream analyses.

As 5'UTR with conserved functions across coronaviruses has been extensively reported (*Chen and Olsthoorn, 2010*, *Madhugiri et al., 2018*, *Yang and Leibowitz, 2015*), we first examined the structure of this region in the PEDV genome for its stability ex vivo. The PEDV 5' UTR structure modeled with SHAPE reactivity constraints contains four stem loops (SL1, SL2, SL4, and SL5; *Figure 1—figure supplement 1C*) similar to other alpha-coronaviruses (*Chen and Olsthoorn, 2010*, *Madhugiri et al., 2018*, *Yang and Leibowitz, 2015*). Our SHAPE dataset showed that 95.6% of highly reactive bases (SHAPE reactivity ≥0.7) are not involved in canonical base pairing or are located adjacent to helical termini or bulges/loops (*Figure 1—figure supplement 1C*). This result suggested that our dataset is reliable as a constraint to predict the intracellular genomic structure of PEDV.

To characterize the intracellular folding features of the PEDV genome, the median SHAPE reactivities and Shannon entropies were calculated in sliding, centered windows (window = 50 nt, step = 1 nt) and a full genome SHAPE structural map was constructed accordingly (*Figure 2*). The global median SHAPE reactivity of the genome is 0.233. Specifically, within the 1–8000 nt region at the 5' end, the median reactivity is 0.173; it increases to 0.268 in the middle segment of the genome (8000–24,000 nt) and then decreases to 0.194 at the 3' end (24,000–28,000 nt). These results indicate that the folding of PEDV genomic RNA within infected cells is nonuniform, with the 5' and 3' ends more compactly structured, while the central region is more flexible (*Figure 2*).

Within the coding regions of the ORF1a, ORF3, M, and N genes, both SHAPE reactivity and Shannon entropy are below the global median, suggesting that these regions are stably folded in the genome (*Figure 2—figure supplement 1*). In contrast, the coding regions of the S and E genes exhibit more relaxed and variable conformations, with both SHAPE reactivity and Shannon entropy exceeding the global median (*Figure 2—figure supplement 1*). Among the known functional regions, the 5' UTR (position: 1–310 nt), which is involved in translation initiation and genome packaging, exhibited lower SHAPE reactivity and lower Shannon entropy compared to the global median, indicating a well-folded overall structure. The FSE (position 12605–12690), which is involved in mediating the ribosomal frame shift, exhibited high SHAPE reactivity and low Shannon entropy values. The 3' UTR (position 27,702–28,044 nt), which is responsible for genome cyclization, mRNA stability, and the initiation of negative-strand synthesis, showed low SHAPE reactivity and high Shannon entropy values. These functional regions in the PEDV genome exhibit distinct SHAPE reactivity, and Shannon entropy profiles suggest that these regions may adopt different structural folding patterns to perform specific functions (*Figure 1*).

### Functional categorization of PEDV genome reveals potentially targetable RNA

To further analyze other functional regions through SHAPE reactivity and Shannon entropy, we combined these two parameters for the classification of structural features (see 'Materials and methods'). Low SHAPE regions were defined as windows in which >75% of the bases exhibited SHAPE reactivities below the global median (calculated in the full PEDV genome), and low Shannon entropy regions were defined according to the same rule, following Manfredonia's method (*Manfredonia et al., 2020*). Windows in which >75% of their bases exhibiting values above the global median were picked as high SHAPE or high Shannon regions. Finally, four categories of RNAs in the PEDV genome were defined according to the relative values of the SHAPE reactivity and Shannon entropy (*Figure 1*, *Supplementary files 2 and 3*).

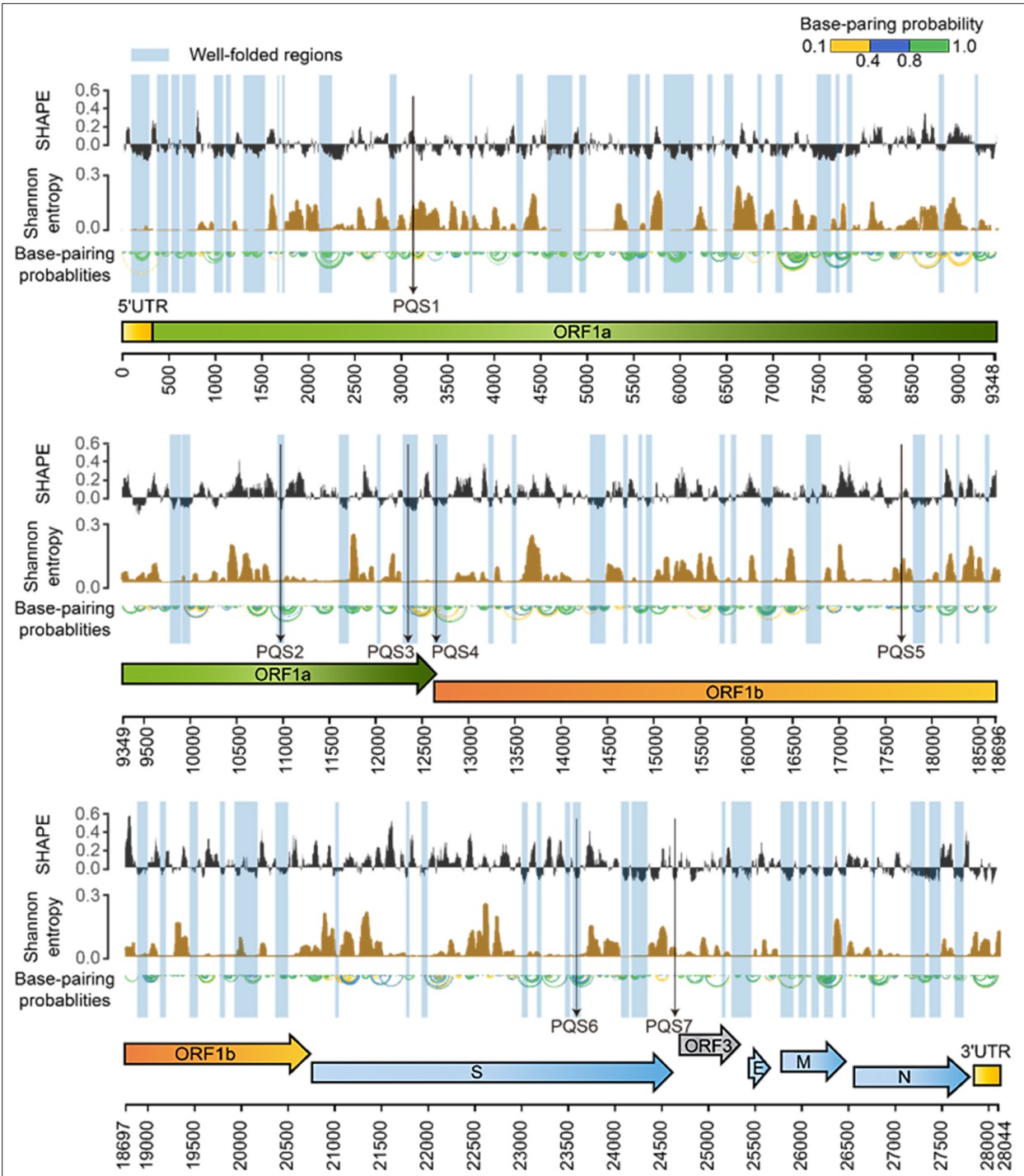

**Figure 2.** SHAPE structural map of the porcine epidemic diarrhea virus (PEDV) genome in infected cells. From top to bottom: SHAPE reactivity, Shannon entropy, base-pairing probabilities, translation reading frames are indicated by arrows, and genomic coordinates are marked at the bottom. Well-folded regions with low SHAPE reactivity and low Shannon entropy are shaded in blue. Potential G4 forming sequences (PQSs) are marked with a black arrow.

The online version of this article includes the following figure supplement(s) for figure 2:

**Figure supplement 1.** Folding characteristics of known functional regions and coding areas in the porcine epidemic diarrhea virus (PEDV) genome.

**Figure supplement 2.** Well-folded structures 1–8 in porcine epidemic diarrhea virus (PEDV) genome.

**Figure supplement 3.** Well-folded structures 9–15 in porcine epidemic diarrhea virus (PEDV) genome.

**Figure supplement 4.** Well-folded structures 10–21 in porcine epidemic diarrhea virus (PEDV) genome.

*Figure 2 continued on next page*

**Figure supplement 12.** Structures of 5'UTR-SL5 in the genomes of porcine epidemic diarrhea virus (PEDV) and SARS-CoV-2.

**Figure supplement 13.** ITC binding profile of (**A**) compound 1 (1-(2-methylpyrimidin-4-yl) piperidin-4-amine; 250 μM) and (**B**) compound 4 (4-(4-chlorophenyl) pyrimidin-2-amine; 250 μM) titrated into the 10 μM 5'UTR-SL5 of porcine epidemic diarrhea virus (PEDV).

Regions with high base-pairing rates (low SHAPE) accounted for 38.10% of the PEDV genome, including the 5' UTR and 3' UTR (*Figure 1B and C*). Of these, the low SHAPE-low Shannon regions (26.40%) maintain stable conformations throughout the viral life cycle and are assumed to represent the well-folded skeleton and the potentially functional regions of the viral genome (*Figure 1B*). Sixty well-folded structures were identified, which exhibited consistent conformations in two replicates (*Figure 2—figure supplements 2–11*). The low SHAPE-high Shannon regions (11.65%), which exhibit variable conformations, might perform different functions through conformational transitions (*Figure 1C*). The design of drugs targeting these regions would require the identification of the specific conformations that are responsible for these functions.

Regions with low basepairing rates (high SHAPE) represented 36.57% of the PEDV genome, including the TRS-Ls of the 5' UTRs (positions 57–72 nt) and FSE regions (positions 12,605–12,690 nt) (*Figure 1D and E*). RNAs in the high SHAPE-high Shannon regions (26.90%) are prone to conformational changes, and their high accessibility suggests that they are more susceptible to interaction with small molecule drugs (*Figure 1D*). In contrast, RNAs in the high SHAPE-low Shannon regions (9.67% of the PEDV genome) maintain a stable single-stranded state within the cell, facilitating pairing with exogenous complementary sequences (*Figure 1E*). Consequently, these regions are considered ideal targets in the design of antiviral oligonucleotides (*Manfredonia et al., 2020*, *Piasecka et al., 2020*, *Sagan et al., 2010*).

## Not all folded functional structures are potential antiviral targets

Although well-folded structures are thought to be potential targets for antiviral molecules (*Childs-Disney et al., 2022*, *Sreeramulu et al., 2021*, *Huston et al., 2021*, *Manfredonia et al., 2020*), there have been relatively limited experimental validations. The 5'UTR-SL5 of coronaviruses is required for defective interfering (DI) RNA replication (*Brown et al., 2007*) and is important for viral packaging (*Masters, 2019*). In the PEDV genome, the 5'UTR-SL5 with a four-way junction structure exhibits well-folded characteristics, similar to those of SARS-CoV-2 (*Figure 2—figure supplement 12*). Seven compounds (*Supplementary file 4*, compounds 1–7), previously reported to interact with 5' UTR-SL5 of SARS-CoV-2 (*Sreeramulu et al., 2021*), were found to bind with the 5' UTR-SL5 of PEDV in vitro, and two compounds (compounds 1 and 4) exhibited dissociation constants (KD) at the μM-level (*Figure 2—figure supplement 13*). However, none of these compounds inhibited the proliferation of PEDV at 50 μM in cells (*Figure 2—figure supplement 14*). This result suggests that compounds bound to well-folded functional structures in vitro do not necessarily exhibit antiviral activity in cells. The accessibility of compounds to potential RNA targets could be an important issue, suggesting that the targetability of the 60 well-folded RNA structures in the PEDV genome requires further validation.

## Identification of PQSs in dynamic single-stranded regions

Dynamic RNAs in the high SHAPE-high Shannon regions can be induced by compounds to form stable structures that perform specific biological functions (*Ratni et al., 2021*). We identified 49 regions with high SHAPE reactivity and high Shannon entropy with at least 25 nucleotides in length (*Supplementary file 3*), spanning a total of 1748 bases (6.2% of the PEDV genome).

To investigate whether RNA motifs in high SHAPE-high Shannon regions can be stabilized to inhibit viral proliferation, we focused on the G4 structure in these regions. Nine putative PQSs were identified by at least three of the G4 prediction tools (*Figure 3A*, *Figure 3—figure supplement 1*). Given the abundance of PEDV mutant strains in nature, further conservation analyses identified seven PQSs that are highly conserved (>90%) among 761 PEDV strains as potential functional G4 candidates (PQS1–PQS7, *Supplementary file 5*). PQS2, PQS3, PQS4, and PQS6 form stable double-stranded structures with low SHAPE reactivity and low Shannon entropy, making them less likely to fold into stable G4 structures (*Figure 3—figure supplements 2–5*). PQS5 and PQS7 did not fall into our four defined regions and were also omitted from further investigation. In contrast, PQS1 is located in the dynamic single-stranded region with high SHAPE reactivity and high Shannon entropy (*Figure 3B and C*), with little competition from local hairpin structure folding.

To validate whether SHAPE analysis could reflect the competitive conformational folding of PQSs in the PEDV genome, we performed in vitro transcription to obtain local intact structures containing PQSs within dynamic single-stranded regions and stable double-stranded regions (*Supplementary file 6*). Thioflavin T (ThT) fluorescence turn-on assays were performed under physiological K+ conditions (100 mM), with the G4 sequence of porcine reproductive and respiratory syndrome virus (PRRSV) serving as a positive control (Control-G4) (*Fang et al., 2023*). The results demonstrated that for short PQSs sequences containing only G4-forming motifs (*Supplementary file 7*), PQS1, PQS3, PQS4, and PQS6 all induced significant enhancement of ThT fluorescence (*Figure 3D and E*, *Figure 3—figure supplement 6*), confirming their ability to form G4 structures. However, in long RNA fragments that encompass PQSs and their flanking sequences (PQSs-long chain), only PQS1 and PQS4 exhibited pronounced ThT fluorescence responses (*Figure 3D and E*), while PQS2, PQS3, and PQS6 showed negligible signals (*Figure 3E*, *Figure 3—figure supplement 6*). Notably, the PQS1-long chain displayed the strongest fluorescence signal, while its mutant counterpart (PQS1mut-long chain) exhibited the lowest background fluorescence (*Figure 3D*). These findings indicate that although most PQSs can form G4 structures in vitro, PQS1—located in the high SHAPE-high Shannon entropy region—demonstrates the most robust G4-forming capability when competing with local secondary structures in the genomic context. Therefore, PQS1 was selected for further structural and functional validation.

PQS1 is located in the nsp3 gene of the PEDV genome (*Figure 3F*) and is highly conserved in the 761 PEDV strains (98.37%, *Figure 3—figure supplement 7A*). Native polyacrylamide gel electrophoresis (PAGE) and circular dichroism (CD) measurements were performed to further verify the formation of the PQS1 G4 structure. The migration rate of PQS1 accelerated noticeably with increasing K+ concentration (*Figure 3—figure supplement 7B*), while the migration rate of mutant sequences (PQS1m) that did not form G4 remained constant. CD spectra revealed that PQS1 adopted a parallel G4 structure in the presence of K+, with the corresponding typical features of a positive CD band at 265 nm and a negative peak near 240 nm (*Figure 3—figure supplement 7C*). 1H nuclear magnetic resonance (NMR) was used to further characterize the G4 structure of PQS1. Chemical shifts in the 10–12 ppm range are considered hallmarks of G4 structures (*Adrian et al., 2012*). The 1H NMR spectrum of PQS1 showed prominent imino proton peaks in this region, confirming the formation of the PQS1 G4 structure (*Figure 4A*).

## Stabilization of PQS1 as a G-quadruplex inhibits RNA synthesis, gene expression, and viral proliferation

The addition of the G4-specific ligand Braco-19 significantly increased the melting temperature (Tm) of PQS1 by 10.8°C (*Figure 4B*), indicating that the RNA G4 structure of PQS1 is stabilized by Braco-19. Furthermore, RNA stop assays and EGFP expression inhibition experiments highlighted the effect of a stable G4 structure of PQS1 in inhibiting RNA replication (*Figure 4—figure supplement 1*) and protein expression (*Figure 4C*, *Figure 4—figure supplement 2*).

To explore whether PQS1 could be a potential drug target, we constructed a recombinant virus without PQS1 (AJ1102-PQS1mut) through a synonymous mutation (G3109A) that does not change the amino acid code (*Figure 4—figure supplement 3A*). The level of AJ1102-PQS1mut proliferation was significantly faster than that of the wild-type strain rAJ1102, suggesting that the disruption of the PQS1 folding facilitates viral proliferation (*Figure 4D*). Another potential G4 sequence, PQS3, located in a stable double-stranded region with low SHAPE reactivity and low Shannon entropy, was used as a

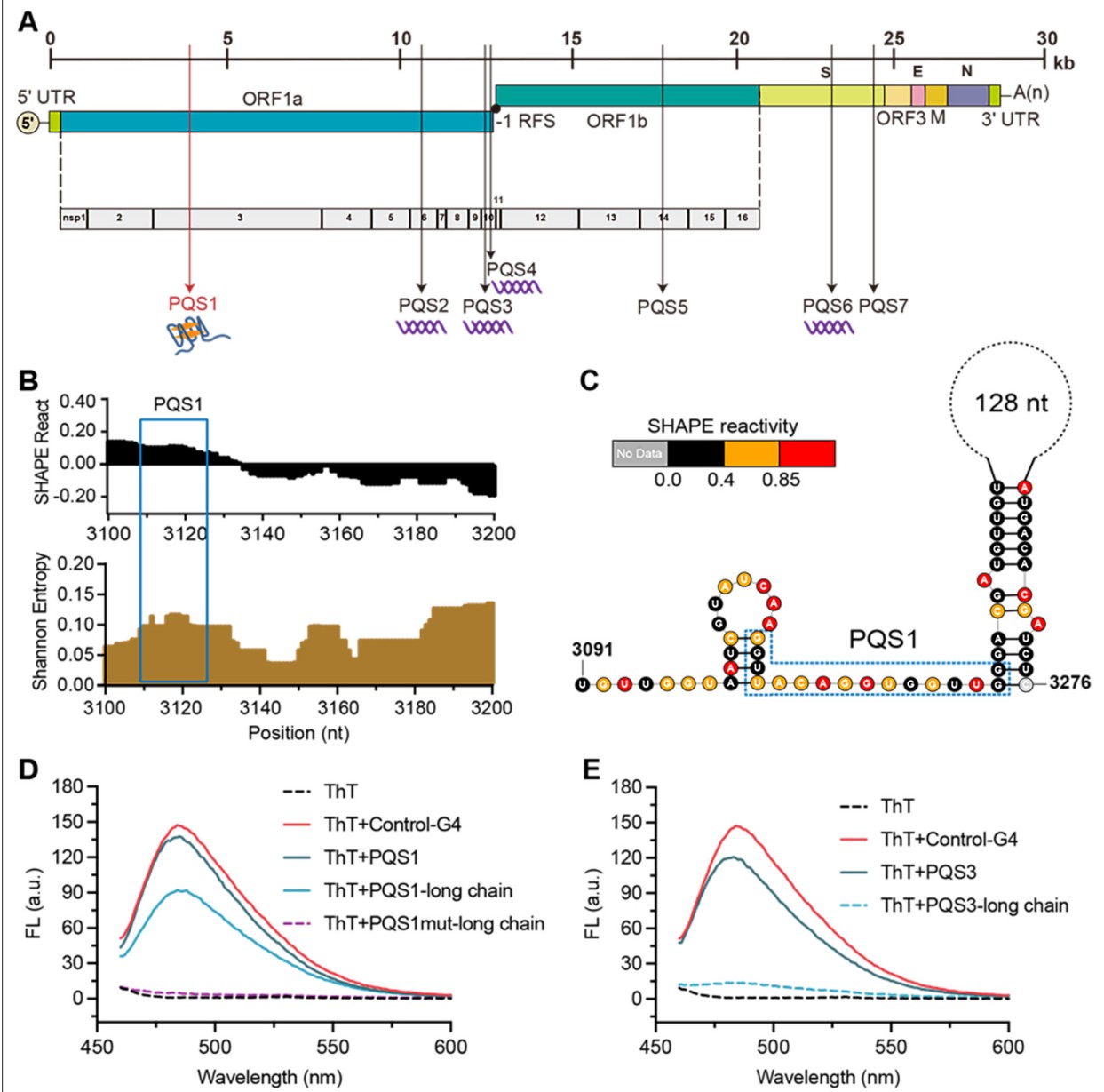

**Figure 3.** The PQS1 within high SHAPE-high Shannon regions has the potential to form G4 structure in the porcine epidemic diarrhea virus (PEDV) genome. (**A**) Distribution of PQSs in the PEDV genome. (**B**) SHAPE reactivity and Shannon entropy for the regions containing PQS1. (**C**) Local secondary structure of the region containing PQS1 predicted with SHAPE reactivity constraints. PQS1 is marked with a blue dashed box. (**D**) Fluorescence turn-on assays of ThT (1 μM) in the presence of PQS1 (0.5 μM), PQS1-long chain (0.5 μM), and PQS1mut-long chain (0.5 μM). (**E**) Fluorescence turn-on assays of ThT (1 μM) in the presence of PQS3 (0.5 μM) and PQS3-long chain (0.5 μM). Excitation wavelength was 442 nm. PRRSV-G4 RNA was used as control.

The online version of this article includes the following source data and figure supplement(s) for figure 3:

**Figure supplement 1.** Prediction of potential G-quadruplex forming sequences (PQSs) in the porcine epidemic diarrhea virus (PEDV) genome by different prediction tools.

**Figure supplement 2.** Secondary structure of the local region containing PQS2 predicted by SHAPE reactivity as a constraint.

**Figure supplement 3.** Secondary structure of the local region containing PQS3 predicted by SHAPE reactivity as a constraint.

**Figure supplement 4.** Secondary structure of the local region containing PQS4 predicted by SHAPE reactivity as a constraint.

**Figure supplement 5.** Secondary structure of the local region containing PQS5 predicted by SHAPE reactivity as a constraint.

**Figure supplement 6.** PQSs in well-folded regions are resistant to G4 formation.

**Figure supplement 7.** G-quadruplex structure folded by PQS1.

*Figure 3 continued on next page*

*Figure 3 continued*

**Figure supplement 7—source data 1.** Original native-PAGE gel for *Figure 3—figure supplement 7B*, indicating the relevant bands and treatments.

**Figure supplement 7—source data 2.** Original files for native-PAGE gel displayed in *Figure 3—figure supplement 7B.*

control. The PQS3 mutant recombinant virus (AJ1102-PQS3mut) proliferated at the same level as the wild type after synonymous mutation (G12322A) of its G4 sequence (*Figure 4—figure supplement 3B*).

Braco-19 inhibited PEDV proliferation with a median effective inhibitory concentration (EC50) of 3.60 μM (*Figure 4—figure supplement 4A*), which is well below the median cytotoxic concentration (CC50 > 100 μM) of Braco-19 (*Figure 4—figure supplement 4B*). In contrast, no inhibition was observed for Braco-19 against the mutant strain AJ1102-PQS1mut (*Figure 4E*). Western blot analysis was used to determine the effects of Braco-19 on viral protein expression in infected cells using the PEDV N protein antibody. The production of the AJ1102-WT N protein was significantly reduced with increasing concentrations of Braco-19, while no change was observed in the production of the

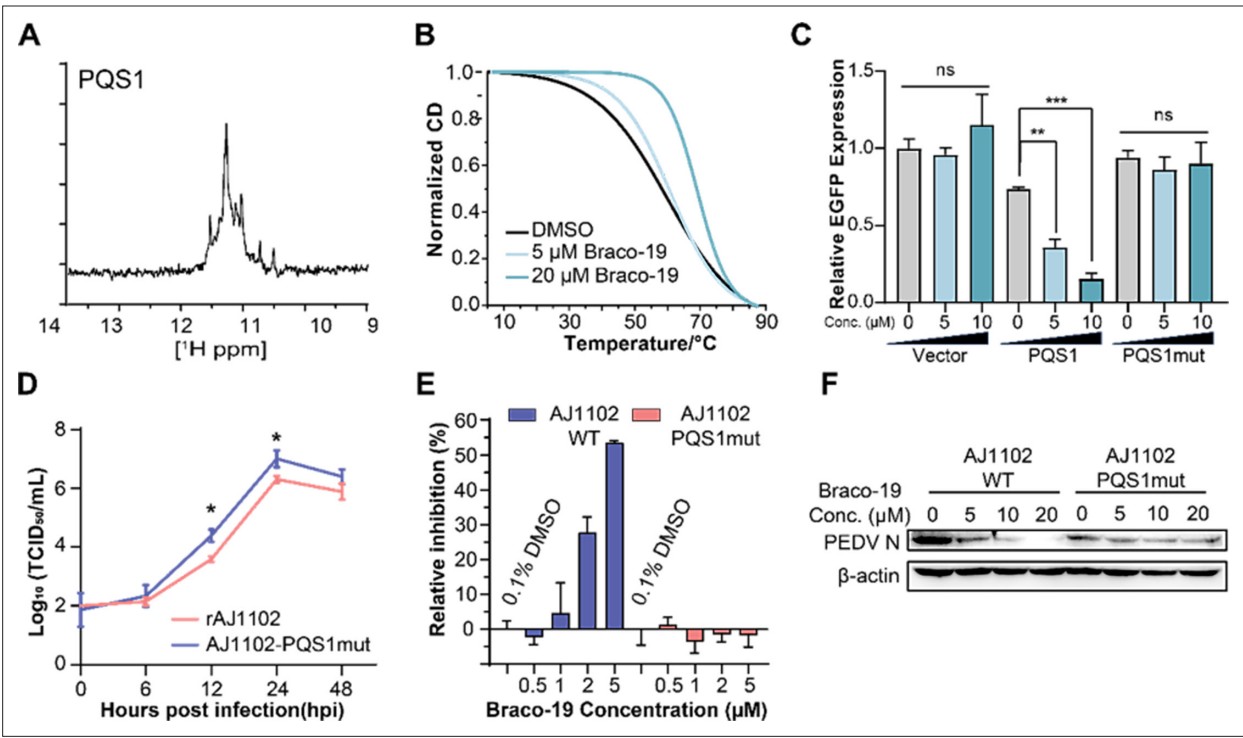

**Figure 4.** The G-quadruplex structure, biological functions of PQS1, and antiviral effects of Braco-19. (**A**) 1H nuclear magnetic resonance (NMR) analysis of PQS1. (**B**) Circular dichroism (CD) melting profiles of PQS1. (**C**) Quantitative fluorescence signal using the corrected total cell fluorescence method for EGFP in cells transfected with plasmids containing the empty vector, PQS1 and PQS1mut. (**D**) Proliferation curve of the PEDV wild type (WT) strain and PQS1 mutant strain. (**E**) The relative inhibition rates of Braco-19 against AJ1102-WT and AJ1102-PQS1mut. (**F**) Western blot analysis of the effects of Braco-19 on the viral N protein expression of AJ1102-WT and AJ1102-PQS1mut.

The online version of this article includes the following source data and figure supplement(s) for figure 4:

**Source data 1.** Original western blots for *Figure 4F*, indicating the relevant bands and treatments.

**Source data 2.** Original files for western blot analysis displayed in *Figure 4F*.

**Figure supplement 1.** G4 structure of PQS1 inhibits RNA-dependent RNA synthesis.

**Figure supplement 1—source data 1.** Original denaturing-PAGE gel for *Figure 4—figure supplement 1B*, indicating the relevant bands and treatments.

**Figure supplement 1—source data 2.** Original files for denaturing-PAGE gel displayed in *Figure 4—figure supplement 1B*.

**Figure supplement 2.** PQS1 inhibits green fluorescent protein (GFP) expression.

**Figure supplement 3.** The PQS3 in the well-folded region lacks biological function.

**Figure supplement 4.** Antiviral activity and cytotoxicity of Braco-19.

AJ1102-PQS1mut N protein (*Figure 4F*), indicating that Braco-19 inhibited the PEDV protein expression in cells by targeting PQS1. Furthermore, as a control, we observed nearly identical inhibitory activity of Braco-19 against both the PQS3 mutant strain (AJ1102-PQS3mut) and the wild-type virus (*Figure 4—figure supplement 3C*), demonstrating the specificity of Braco-19's action on PQS1.

## Efficient antiviral siRNAs design targeting stable single-stranded regions

High SHAPE-low Shannon regions represent stable, accessible RNAs in cells, exposing long Watson–Crick edges of multiple residues available for base-pairing with complementary sequences (*Figure 1D*). Our analysis of fold characterization (see 'Materials and methods') identified 73 such regions (*Supplementary file 3*) with at least 25 nucleotides in length, spanning a total of 2696 bases (9.6% of the PEDV genome). To test the suitability of these regions as targets for antiviral siRNAs, we designed four siRNAs targeting these regions that (i) have at least 10 consecutive unpaired nucleotides and (ii) were highly conserved in the different PEDV strains (*Supplementary file 7*, *Figure 5A–D*). As a control, we designed four additional highly conserved siRNAs targeting duplex regions (*Figure 5—figure supplement 1*) and one non-targeting siRNA that does not target any sequence of the viral and host genomes (*Supplementary file 7*).

The results of RT–qPCR of infected cells 16 h after viral exposure revealed that four siRNAs targeting high SHAPE-low Shannon regions significantly inhibited viral proliferation in cells (*Figure 5E*) compared to the non-targeting siRNA control (siRNA-NC). Consistent with the RT-qPCR results, compared to siRNA-NC, we observed that these four siRNAs had minimal cytopathic effects (CPEs) in Vero cells (*Figure 5—figure supplement 2*). Further TCID50 analysis demonstrated that three of the four tested siRNAs targeting single-stranded regions reduced the TCID50 by 1.5 log10 titer compared to siRNA-NC (ss-1, ss-2, and ss-3; *Figure 5F*). In contrast, only one of the four siRNAs targeting duplexes could slightly prevent PEDV proliferation in cells (ds-4; *Figure 5E and F*, *Figure 5—figure supplement 2*). The target sequences of four single-stranded targeting siRNAs are more than 85% conserved in 761 known PEDV strains. The combination of two or more of these siRNAs is expected to be effective in inhibiting nearly all known PEDV strains. These results showed that siRNAs targeting single-stranded regions with high SHAPE reactivity and low Shannon entropy can be more effective in exerting their antiviral effects.

## Discussion

Several RNA-targeting antiviral strategies have highlighted the potential for therapeutic application (*Dai et al., 2024*). However, it is challenging to identify these RNAs from the viral genome in situ. In SHAPE-MaP analysis, regions with low reactivity and low entropy values are more likely to form a single stable structure in cells and are therefore considered candidates for potential functional RNAs and targets for antiviral therapy. Novel well-defined RNA structures that are critical for viral fitness in HIV, HCV, DENV, and SINV have been identified by the analysis of low SHAPE and low entropy regions (*Dethoff et al., 2018*, *Kutchko et al., 2018*, *Siegfried et al., 2014*, *Mauger et al., 2015*). However, there are still no clinically approved small molecule drugs or oligonucleotides targeting stable folded RNA structures. Our data show that two compounds interacted with the well-folded 5'UTR-SL5 of PEDV in vitro at μM-level KD, but did not show antiviral activity at 50 μM in cells (*Figure 2—figure supplements 13 and 14*). This result suggests that, on the one hand, the low accessibility of the regions with low SHAPE reactivity makes them difficult to access by compounds in cells and may require compounds to have a better KD with RNA. However, the binding of compounds to the well-folded structure does not necessarily alter its function. In contrast, the FDA-approved compound Risdiplam exerts its biological activity by stabilizing transient RNA structures (*Campagne et al., 2019*, *Ratni et al., 2021*, *Sheridan, 2021*), suggesting that it may be possible to inhibit viral proliferation by stabilizing variable conformations. We therefore focused our attention on highly accessible regions with high SHAPE reactivity in cells.

We identified 42 low SHAPE-high Shannon regions and 49 high SHAPE-high Shannon regions in the PEDV genome by SHAPE-MaP. These regions represent conformationally variable RNAs in the viral genome. As the low SHAPE regions are highly structured, it is difficult for the G4 in these regions to outcompete the folding of long base-paired structures (*Figure 6A*). Therefore, we selected a

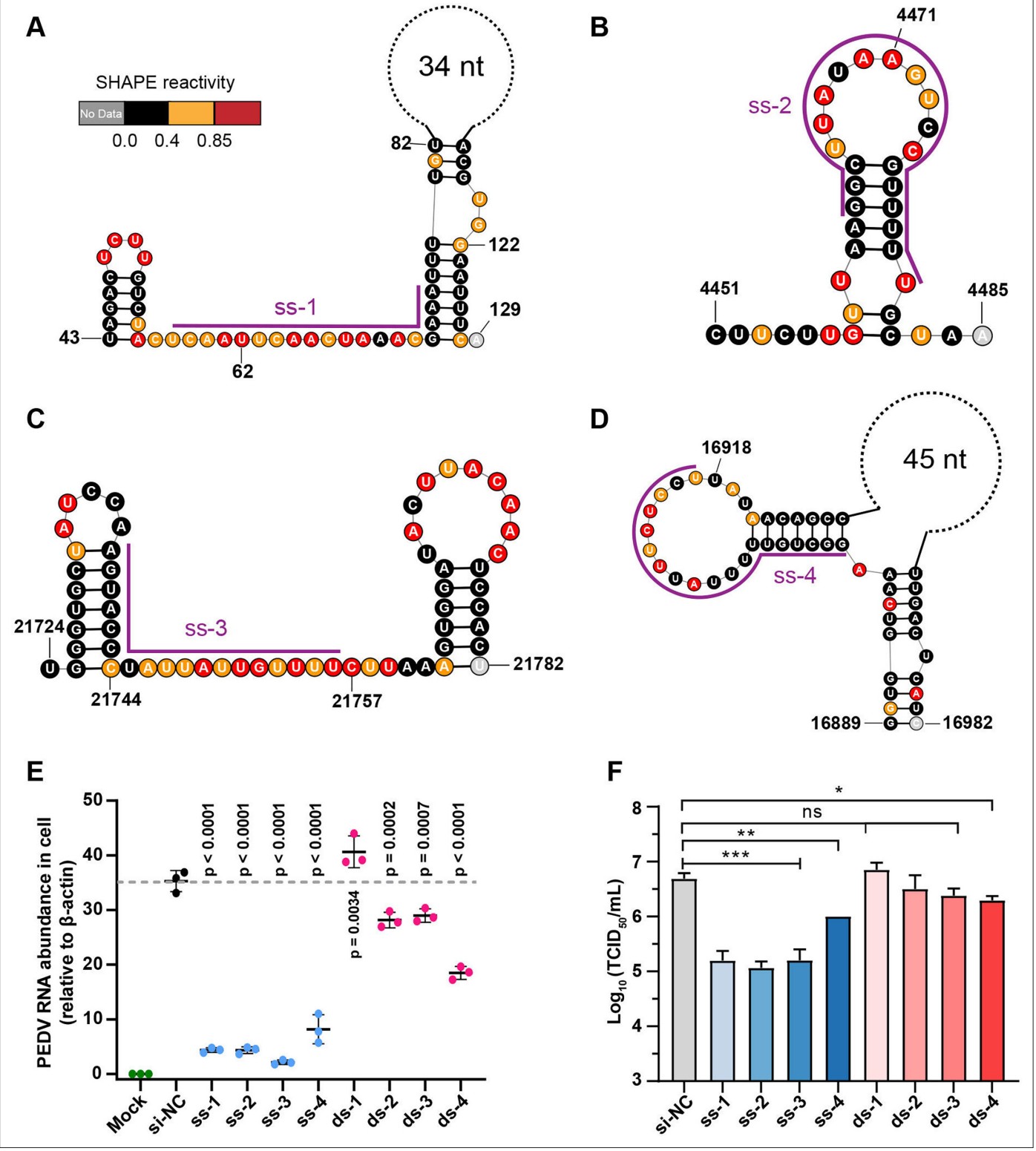

**Figure 5.** Secondary structure of target regions and antiviral effects of siRNAs. (**A–D**) Local secondary structures of the stable single-stranded regions targeted by siRNAs predicted by SHAPE reactivity as a constraint; (**A**) ss-1; (**B**) ss-2; (**C**) ss-3; (**D**) ss-4. (**E**) qPCR showing the relative abundance of the porcine epidemic diarrhea virus (PEDV) RNA genome in infected Vero cells. The four siRNAs targeting the high SHAPE-low Shannon regions and the four siRNAs targeting the duplex regions are labeled ss-1 to ss-4 and ds-1 to ds-4, respectively. ss (single-stranded targeting siRNAs); ds (dual-stranded targeting siRNAs). si-NC was a control siRNA that did not target any viral or host sequences, and the mock group was not inoculated with

*Figure 5 continued on next page*

*Figure 5 continued*

virus. (**F**) TCID50 assays for detecting virus titers. The presented results represent the means and standard deviations of data from three independent experiments. ns: no significant difference. *p<0.05; **p<0.01; ***p<0.001, Duncan's multiple comparison test.

The online version of this article includes the following figure supplement(s) for figure 5:

**Figure supplement 1.** Local secondary structures of the duplex regions targeted by siRNAs.

**Figure supplement 2.** Cytopathic effects (CPE) of siRNA targeting regions of duplex or single-strand on porcine epidemic diarrhea virus (PEDV)-infected Vero cells.

potential G4 sequence (PQS1) located in the high SHAPE-high Shannon region of nsp3 (PLpro), where a G4-disrupting mutation promotes viral proliferation. We found that Braco-19 significantly stabilized the G4 structure of PQS1 in vitro and inhibited PEDV proliferation in cells. Crucially, Braco-19 showed no inhibitory activity against the PQS1-mutant strain while maintaining potent activity against the PQS3-mutant strain (*Figure 4E*, *Figure 4—figure supplement 3C*). This suggests that the compound can selectively target the PQS1 of the high SHAPE-high Shannon region in cells. This approach can be extended to other RNA viruses and potentially to DNA viruses where RNA intermediates play critical roles in the viral life cycle.

Recently, the FDA has approved several siRNAs for therapeutic purposes (*Ali Zaidi et al., 2023*). siRNA-mediated RNAi is particularly suited to inhibit viruses with single-stranded RNA genomes due to its potential to specifically silence virtually any therapeutic target. When screening siRNA target sequences, sequence conservation and functional importance of the target region are usually given priority. However, the RNA structure of the target region also has a significant impact on the efficiency of RNAi (*Figure 6B*; *Cao et al., 2021, Manfredonia et al., 2020, Piasecka et al., 2020, Patzel et al.,*

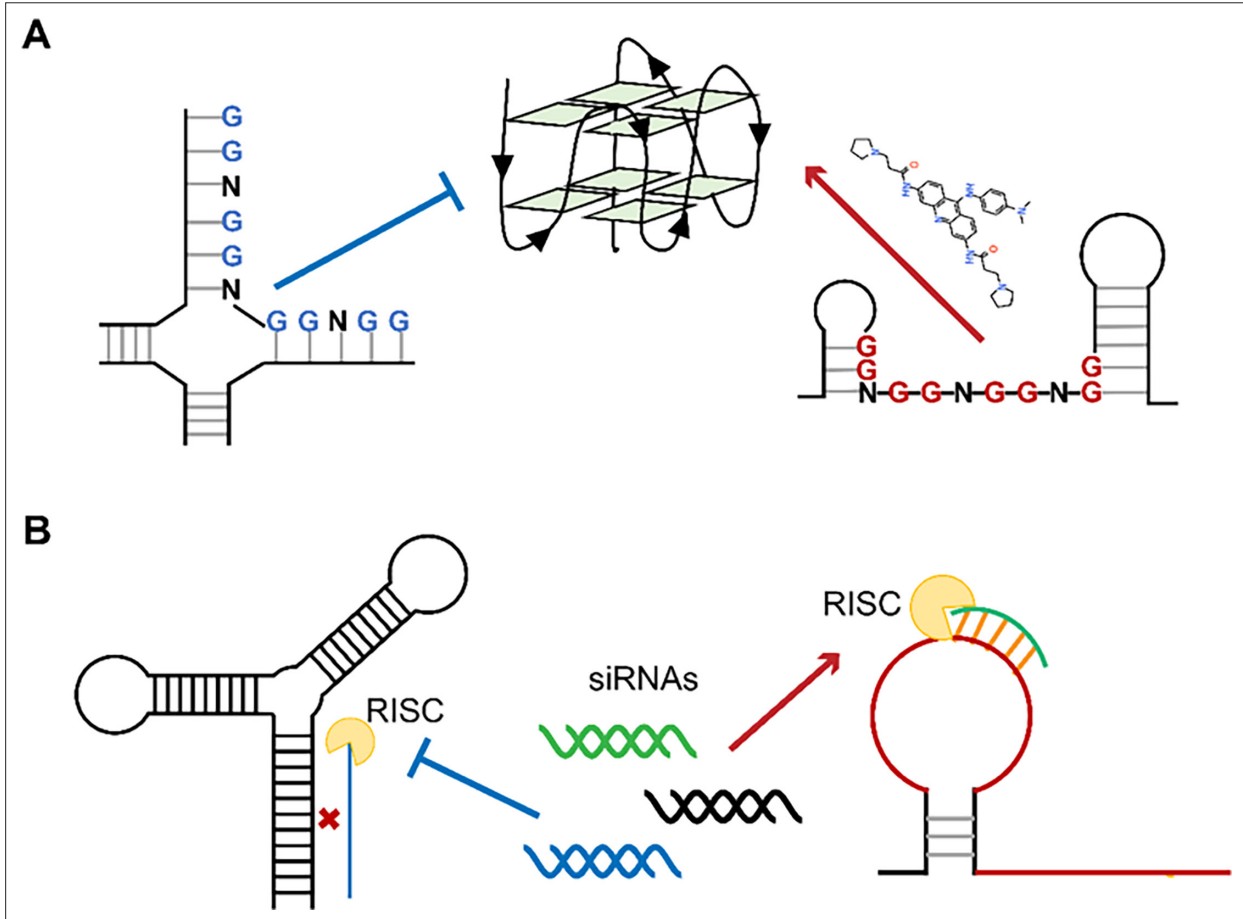

**Figure 6.** Single-stranded RNA regions in viral genomes as the targets for antiviral therapy. (**A**) PQSs in the single strands are easier to be induced into G4s by ligands than those in the paired regions. (**B**) Influence of the secondary structures on the binding efficiency of siRNAs.

*2005*, *Sagan et al., 2010*, *Szabat et al., 2020*). Our approach uniquely integrates in situ RNA structural data (SHAPE reactivity and Shannon entropy) to prioritize siRNA targets within stable single-stranded regions (high SHAPE reactivity, low Shannon entropy), which are experimentally validated as accessible in infected cells. This represents a significant departure from traditional siRNA design methods that rely primarily on sequence conservation, thermodynamic rules (e.g., Tuschl rules), or in vitro structural predictions (*Ali Zaidi et al., 2023*, *Qureshi et al., 2018*, *Tang and Khvorova, 2024*), which may not accurately reflect intracellular RNA accessibility. Bowden-Reid et al. designed 39 antiviral siRNAs against various SARS-CoV-2 variants based on sequence conservation, ultimately identifying 8 highly effective sequences (*Bowden-Reid et al., 2023*). Notably, five of these effective sequences targeted regions that were located in high SHAPE-high Shannon regions according to SARS-CoV-2 SHAPE datasets (*Supplementary file 8*; *Manfredonia et al., 2020*). This independent finding aligns perfectly with our conclusions and demonstrates that SHAPE-based siRNA design outperforms sequence/structure-agnostic approaches, at least in terms of significantly improving antiviral siRNA screening efficiency. Given the growing availability of SHAPE datasets for numerous viruses, we are confident that our methodology will facilitate a more precise design of antiviral siRNAs.

## Materials and methods
### Cell culture and PEDV infection
Vero E6 cells were cultured in 10 cm culture plates in Dulbecco's modified Eagle's medium (DMEM) supplemented with 10% fetal bovine serum at 37°C with 5% $CO_2$. Cells were infected with PEDV strain AJ1102 (GenBank: MK584552.1) at an MOI of 0.1 when the cell confluence reached 95%.

### In situ RNA probing and purification
During the logarithmic phase of viral genome replication after infection (12 h), infected cells were washed three times with 10 mL cold PBS and then resuspended in PBS containing 2-methylnicotinic acid imidazolide (NAI) (*Spitale et al., 2013*) at a final concentration of 100 mM. Two independent replicates were performed for the ex vivo experiments. An equal amount of DMSO was added to another sample as a control. The samples were then incubated at room temperature with continuous shaking for 15 min, after which the reaction was terminated by adding DTT at a final concentration of 0.5 M. The supernatant was then removed and the cells were lysed by adding 6 mL TRIzol reagent. RNA was extracted by adding 1.2 mL of chloroform:isoamyl alcohol (24:1). 3 mL of the upper aqueous phase was transferred to a new tube, 6 mL of anhydrous ethanol was added, and the mixture was incubated overnight at –20°C. RNA was pelleted at 18,000 × $g$ for 20 min at 4°C, washed twice with 80% EtOH, and then the pellet was air-dried. Finally, RNA was dissolved in RNase-free water and then stored at –80°C until further use.

### Isolation and fragmentation of poly(A) PEDV genomic RNA
We isolated poly(A) RNA with VAHTS mRNA capture beads (Vazyme #N401) according to the manufacturer's instructions. 5 µg of total RNA was diluted to 50 µL with RNase-free water and purified in two rounds with mRNA capture beads. Poly(A) RNA was fragmented to a median size of 150–200 nt by incubation at 94°C for 8 min in fragmentation buffer (65 mM Tris–HCl; 4 mM $MgCl_2$; 95 mM KCl; pH 8.0) and then purified with an RNA Clean & Concentrator-5 kit (Zymo Research, R1015) following the manufacturer's instructions.

### Reverse transcription of modified PEDV genomic RNAs
Aliquots of 20 µL poly(A) RNA (~500 ng) were distributed into 0.65 mL RNase-free tubes, followed by the addition of 2 µL random primer (200 ng/µL) to each tube. The samples were incubated at 65°C for 5 min. Sixteen microliters of 2.5×MaP buffer (125 mM Tris, pH 8.0; 187.5 mM KCl; 15 mM $MnCl_2$; 25 mM DTT and 1.25 mM dNTPs) were added to each tube, and the mixture was incubated at 25°C for 2 min. Then, 2 µL SuperScript II reverse transcriptase (200 U/µL; Life Technologies, cat. no. 18064-014) was added and the mixture was incubated at 42°C for 3 h. The reactions were heat-inactivated by incubation at 75°C for 15 min, after which the mixture was placed on ice. cDNA was purified from reactions using G-25 micro spin columns (GE Healthcare, cat. no. 27-5325-01) following the manufacturer's instructions. The second-strand cDNA synthesis reaction was then immediately performed. The

volume of first-strand cDNA was adjusted to 56 µL with nuclease-free $H_2O$, 6.5 µL NEBNext Second Strand Synthesis Reaction Buffer (10×; New England Biolabs, cat. no. E6111S) and 3.5 µL NEBNext Second Strand Synthesis Enzyme Mix (New England Biolabs, cat. no. E6111S) were added and the mixture was then incubated at 16°C for 2.5 h.

## SHAPE-MaP sequencing library construction and quality control

Sequencing libraries were generated using a VAHTS Universal V8 RNA-seq Library Prep Kit for MGI (Vazyme #NRM605) according to the manufacturer's protocol. Libraries concentrations were measured using a Qubit dsDNA HS Assay Kit (Thermo Fisher, cat. no. Q32851). 1 µL of the final library product was removed and the library quality was analyzed using an Agilent DNA1000 Kit (Agilent, cat. no. 5067-1504).

## SHAPE-MaP data analysis

SHAPE-MaP data analysis was adapted from *Manfredonia et al., 2020*, and performed using the RNA Framework v2.6.9 (*Incarnato et al., 2018*; *Incarnato and Brewer, 2025*), employing the following steps:

(a) Data was preprocessed and aligned to the PEDV reference genome using the rf-map module with the parameters: -ctn -cmn 0 -cqo -cq5 20 -b2. The reads with internal Ns were discarded, and low-quality bases (Phred < 20) were excluded before alignment.

(b) The PEDV genome index was built using Bowtie2 software (*Langmead and Salzberg, 2012*). Individual nucleotide mutation signals were assessed with the rf-count module with parameters: -m -rd, generating a binary RC file with sequence data, mutation counts, coverage, and total aligned fragments.

(c) The rf-norm module was applied for data normalization with parameters: -sm 3 -nm 3 -rb AC -n 500 -mm 1, utilizing the scoring method developed by *Siegfried et al., 2014*.

## PEDV RNA secondary structure modeling guided by SHAPE data

The rf-fold module, employing a previously established windowed approach, (*Siegfried et al., 2014*), predicts the RNA secondary structure of the entire PEDV genome using the normalization step's XML file, with a default pseudo-free energy intercept and slope values of –0.6 (kcal/mol) and 1.8 (kcal/mol), respectively, with the following parameters: -fw 3000 -fo 300 -wt 200 -pw 1000 -po 250 -dp -sh -nlp -md 600. This means that RNA structure will generate a 3000 nt window and slide in 300 nt increments over the genome in 1000 nt partition windows, predicting the most reasonable secondary structure of the genome under the condition that the maximum base pairing distance does not exceed 600 nt. SHAPE reactivities were incorporated in the form of soft constraints using the Vienna RNA package (*Lorenz et al., 2016*). For each window, the first and last 100 nt were ignored to avoid terminal biases. Additionally, RF Fold can provide pairing probabilities between bases, Shannon entropy of individual bases, and potential pseudoknot structures. The Shannon entropy is commonly used to measure the diversity of RNA secondary structures:

$$H_i = -\sum_{j=1}^{J} P_{ij} log_{10} P_{ij} \tag{1}$$

where *P* represents the pairing probability between bases (*Equation 1*).

## Identification of characterized regions in viral genomes

Based on previous research methods to identify well-folded regions (*Siegfried et al., 2014*), modifications have been made as follows:

(a) For the identification of low SHAPE-low Shannon regions, we first calculated the local medians of SHAPE reactivities and Shannon entropies in sliding, centered, 50 nt windows, and subtracted the global median for visualization. Then, a 50 nt window was slid along the genome with a 1 nt step. Windows with >75% of the bases below both the global SHAPE and the Shannon median (calculated on the full PEDV genome) were selected, and windows less than 10 nt apart were merged. Following the initial identification of well-folded regions, entire structures exhibiting consistent conformations in both datasets and having at least 50% of their

bases in well-folded regions were further selected. Subsequently, the well-folded RNA structures within cells were determined.

(b) For the identification of low SHAPE-high Shannon regions, a 50 nt window was slid along the PEDV genome. Windows with >75% of the bases below the global SHAPE median and above the global Shannon median (based on the full PEDV genome) were selected, and windows less than 10 nt apart were merged.

(c) For the identification of high SHAPE-low Shannon regions, instead, a window of 25 nt was slid along the PEDV genome. Windows with >75% of the bases were above the global SHAPE median and below the global Shannon median (calculated on the full PEDV genome), and >50% of the bases were predicted to be single-stranded in the MEA structure; windows less than 10 nt apart were selected and merged.

(d) For the identification of high SHAPE-high Shannon regions, a 25 nt window was slid along the PEDV genome. Windows with >75% of the bases above both the global SHAPE and Shannon median (based on the complete PEDV genome) were selected, and windows less than 10 nt apart were merged.

## G-quadruplex forming sequence prediction and conservativeness analysis

The G4 sequence was defined as $G_{\geq 2} N_{1-6} G_{\geq 2} N_{1-6} G_{\geq 2} N_{1-6} G_{\geq 2}$, where $G$ refers to guanine and N refers to any nucleotide, including guanine. Four independent G4 prediction tools (QGRS-mapper, G4Catchall, ImGQfinder, and pqsfinder) were used to analyse the potential G4 forming sequences (PQSs) in the PEDV genome (*Doluca, 2019*, *Hon et al., 2017*, *Kikin et al., 2006*, *Varizhuk et al., 2017*). Conservation analysis was performed using Molecular Evolutionary Genetics Analysis (MEGA) software (version 6.0; available at https://www.megasoftware.net/). 761 PEDV genome sequences were retrieved from the National Center for Biotechnology Information website (NCBI, https://www.ncbi.nlm.nih.gov/). The graphical representation of the alignment sequence was generated by WebLogo 2 (https://weblogo.berkeley.edu/) (*Crooks et al., 2004*).

## Verification of G-quadruplex structure formation by native PAGE

Cy2-tagged RNA was annealed in buffers (100 mM potassium arsenate, pH 7.0) containing different concentrations of KCl (0, 50 mM, 100 mM). The marker used in this experiment was a mixture of P15 (UAAUACGACUCACUA), M17 (UAAUACGACUCACUAUAUA), and M20 (UAAUACGACUCACUAUACGA). Acrylamide at a 20% concentration was used to separate the gel. The concentration of the RNA sample was 100 nM and electrophoresis was performed at 100 V for 2 h. Native PAGE was carried out under a controlled temperature (4.0°C) with a vertical electrophoresis instrument (DYCZ-22A; Bio-Rad, Beijing, China). After electrophoresis, RNA oligomers were scanned using a Pharos FX molecular imager for visualization.

## CD spectroscopy

RNAs (20.0 µM) were dissolved in 10 mM Tris–HCl buffer (pH 7.0) containing varying concentrations of KCl (0, 50, 100 mM). The CD experiment was carried out at 25°C using a JASCO-810 spectrophotometer (Jasco, Easton, MD, USA) with a quartz cell path length of 1.0 mm. CD spectra were collected from 320 to 200 nm with a scanning speed of 50 nm/min. All CD spectra were baseline corrected for the signal contribution of the buffer and represent the mean of at least two repeats.

## NMR spectroscopy

1H NMR spectra were recorded at 298 K using an 800-MHz Brock Avance DRX-800 spectrometer equipped with a low-temperature triple-resonance reverse autotuning and matching probe. The presumption of water is achieved by excitation engraving. RNA samples were dissolved in DEPC solution containing 200 mM KCl and 10% $D_2O$ at a final concentration of 0.5 mM. The sample was heated at 95°C for 5.0 min and slowly cooled to 25°C. The facility was set for 100 ms diffusion time (TD), 50 ms eddy recovery time (TE), and 2s relaxation delay (TR).

## CD melting studies

Braco-19 was dissolved in DMSO at a concentration of 20 mM. RNA samples were dissolved in buffer containing 10 mM Tris–HCl (pH 7.0) and 100 mM KCl at a final concentration of 20 µM. The samples

were heated at 95°C for 10 min and then slowly annealed to room temperature. CD melting was measured on a Jasco-810 spectrophotometer equipped with a water bath temperature control attachment. CD melting curves were recorded using a heating rate of 0.2°C/min and absorbance values were collected at 1°C intervals. The thermal stability of G4 RNA was measured by recording the CD value at 264 nm in relation to temperature. The required temperature range was set from 4.0°C to 98°C.

## RNA stop assay

P15 (300 nM) and template RNA (600 nM) were annealed in buffer at 25°C (50 mM HEPES, 20 mM NaCl, 1.0 mM KCl, 5 mM MgCl$_2$, and 4 mM DTT [pH 7.0]), heated at 95°C for 5.0 min, and slowly cooled to 4°C. Subsequently, NTPs (final concentration, 200 µM) and 3Dpol (recombinant RNA-dependent RNA polymerase; Abcam, ab277617, 0.02 mg/reaction) were added to the system, and the reaction was carried out at 33°C for 20 min. An equal amount of stop buffer (95% formamide) was added, and the reaction was stopped after heating at 90°C for 4.0 min. The products were loaded and separated on a 20% denatured polyacrylamide gel. Finally, the gel was scanned using a Pharos FX molecular Imager (Bio-Rad) operated in fluorescence mode.

## EGFP expression repression by RNA G-quadruplex stabilization

We constructed pEGFP-C1 vector with wild-type PQS1 sequences or G4-mutated PQS1mut sequences fused to the N-terminus of an EGFP reporter gene. Vero cells were transfected with Lipofectamine 3000 (Invitrogen, USA) according to the manufacturer's protocol and incubated in media supplemented with various concentrations of Braco-19 (*Gowan et al., 2002*) for 48 h. The cells were then washed with PBS and fixed at 25°C for 15 min with 4% paraformaldehyde. The samples were washed three times with PBS for 5 min each and then fixed with precooled methanol at –20°C for 10 min. Observations were conducted using a two-photon laser scanning confocal microscope (Olympus, fv1000mp).

## Construction of the RNA G4-mutated recombinant virus

The pBAC-CMV-PEDV template was obtained from Prof. Xiao Shaobo (College of Veterinary Medicine, Huazhong Agricultural University). The primers used were chemically synthesized (*Supplementary file 1*). The G-rich region sequence (3109–3125 nt) was mutated (AAGGTTACAGGTGGTTGG to AAAGTTACAGGTGGTTGG). Two pairs of primers (PQS1Mut-F1 and PQS1Mut-R1, PQS1Mut-F2 and PQS1Mut-R2) were used to amplify the fragment containing the mutant site. The PQS1mut fragment containing the mutation sites was amplified by overlap PCR. A scaffold oligo was used as a template, PQS1m-sgRNA-a and PQS1m-sgRNA-a were used as primers to synthesize guide DNA, and then guide RNA (sgRNA) was obtained by reverse transcription. pBAC-CMV-PEDV was cut with the Cas9 enzyme and sgRNA. The PQS1mut fragment was cloned and inserted into a linearized pBAC-CMV-PEDV vector to generate the pBAC-PEDV-PQS1mut mutant plasmid. Vero cells were placed on 6-well plates and plasmid transfection was performed when the cells grew to 60% to 80% confluence according to the manufacturer's protocol (Lipofectamine 3000, Thermo, L3000015). If no lesions appeared, the freeze–thaw solution was added to the six-well plate with 80% confluence after three repeated freeze–thaw cycles for further culture. When lesions appeared, the successful mutation of the mutation site was confirmed by reverse transcription sequencing and the PEDV-PQS1mut mutant strain was obtained.

## Determination of the antiviral activity of Braco-19 against PEDV by RT-qPCR

Braco-19 at different concentrations (0.5, 1, 2, and 5 µM) were inoculated into monolayer Vero cells and incubated for 2 h. PEDV-WT and PEDV-PQS1mut were inoculated at an MOI (multiplicity of infection) of 0.1 and incubated for 2 h. DMEM was replaced with DMEM containing Braco-19, and the mixture was incubated for 16 h. cDNA was synthesized from 100 ng of total RNA per sample using the Transcriptor First Strand cDNA Synthesis Kit (Roche, Switzerland) according to the manufacturer's instructions. Quantitative PCR (qPCR) was carried out by following the Minimum Information for Publication of Quantitative Real-Time PCR Experiments (MIQE) guidelines (*Bustin et al., 2009*) using Taq Pro Universal SYBR qPCR Master Mix (Vazyme, Q712) according to the manufacturer's protocol.

mRNA expression was normalized to beta-actin levels with primers beta-actin-F/R (*Supplementary file 1*). PEDV genomic RNA was quantified at the nonstructural protein 12 (nsp12) gene using primers qPCR-nsp12-F/R (*Supplementary file 1*).

### Western blot analysis to detect PEDV N gene expression

Protein samples were separated by SDS–PAGE using a 12.5% polyacrylamide gel and then transferred to a polyvinylidene difluoride (PVDF) membrane (Millipore, USA). The membrane was blocked with 5% BSA in Tween-PBS buffer and incubated overnight at 4°C with primary antibodies. After three washes, it was incubated with HRP-conjugated secondary antibodies (Beyotime, China). Following three more washes, protein bands were detected using enhanced chemiluminescence (Bio-Rad, USA).

### CCK-8 assays

Vero cells were treated with graded compound concentrations for 12 h in 96-well plates, then incubated with 10 µL CCK-8 reagent (Beyotime) for 2 h at 37°C. Absorbance at 450 nm was measured using a microplate reader to assess proliferation.

### Indirect immunofluorescence assay (IFA)

Cells in 24-well plates were pretreated with test compounds, infected with PEDV (MOI=0.1, 2 h, 37°C), then maintained in compound-containing DMEM. After 24 hpi, cells were fixed, permeabilized, and washed before blocking and incubating with primary (mouse anti-PEDV N protein antibody) and Alexa Fluor 488-secondary antibody. Nuclei were DAPI-stained, and fluorescence images were captured using an Olympus IX73 microscope.

### Isothermal titration calorimetry (ITC)

ITC experiments were carried out on an AutoiTC100 titration calorimeter (MicroCal) at 25°C. All solutions (buffer, RNA, and ligand) were degassed prior to experimentation. A 10 µM solution of RNA 5' UTR-SL5 was introduced into the cell and a compound solution (250 µM) was added to the rotating syringe (750 rpm). The RNA solution received 40 µL of compound through 20 sequential injections (1.5 min intervals), preceded by a 0.4 µL priming injection to compensate for syringe diffusion during system equilibration. This initial injection was excluded from the data fitting. Furthermore, dilution experiments were performed with buffer (100 mM HEPES [pH 8.0], 100 mM NaCl, and 10 mM $MgCl_2$) in the cell, and the compound remained in the syringe. We utilized Microcal ITC analysis software to analyze the raw ITC data using the two-site binding model. The dilution data were subtracted from the raw interaction data of the compound with RNA prior to analysis. Following fitting of these experimental data, the enthalpy change during the process was determined.

### siRNA antiviral assay

siRNAs were synthesized in GENERAL BIOL and transfected into $8×10^4$ Vero E6 cells in 24-well plates using 10 pmol of each siRNA and Lipofectamine RNAiMAX Transfection Reagent (Invitrogen, 13778075) according to the manufacturer's protocol. Transfected cells were infected with PEDV (MOI = 0.1) after 24 h. After 16 h of PEDV infection, minimal CPEs on Vero cells were observed under the microscope. To quantify PEDV RNA levels in infected cells, total RNAs were extracted with TRIzol Reagent (Invitrogen, 15596026). For RT-qPCR, 100 ng of total RNAs were converted into cDNA using HiScript II 1st Strand cDNA Synthesis Kit (Vazyme, R212). qPCR was performed with ChamQ Universal SYBR qPCR Master Mix (Vazyme, Q711) on a StepOnePlus real-time PCR machine (Applied Biosystems). Primers targeting the nucleocapsid (N) gene of PEDV and the beta-actin gene of Vero cells were used for qPCR. For the 50% tissue culture infectious dose (TCID50) assay, monolayers of Vero cells seeded in 96-well plates were washed twice with DMEM. The supernatants of infected cells were serially diluted tenfold. Cells were then inoculated with eight replicates of the appropriate dilutions of the virus suspension. TCID50 was calculated using the Reed-Muench method after culturing the cells for 48 h at 37°C in 5% $CO_2$.

### Statistical analysis

The experimental data was analyzed by Student's *t*-test or one-way or two-way ANOVA using GraphPad Prism 8 software. *$p<0.05$ was considered to indicate statistical significance.

## Acknowledgements

This work was supported by the National Key R&D Plan of China (2021YFD1801102), the Fundamental Research Funds for the Central Universities (2662023PY005), the National Natural Science Foundation of China (32473010), the China Postdoctoral Science Foundation (2024M761080), the Natural Science Foundation of Hubei Province (2025AFB288), the HZAU-AGIS Cooperation Fund (SZYJY2022016 to DW), the National Natural Science Foundation of China (22077043), the Natural Science Foundation of Hubei Province (2021CFA061 to DW), and funding from the State Key Laboratory of Agricultural Microbiology (AML2023B05 to DW). The funders had no role in the study design, data collection and analysis, decision to publish, or preparation of the manuscript. We thank Professor Shaobo Xiao (Huazhong Agricultural University) for providing the PEDV AJ1102 strain and for his guidance in virological experiments. The high-throughput sequencing, ITC, and confocal microscopy data were acquired at the State Key Laboratory of Agricultural Microbiology Core Facility.

## Additional information

### Competing interests

Shozeb Haider: Reviewing editor, eLife. The other authors declare that no competing interests exist.

### Funding

| Funder | Grant reference number | Author |
| --- | --- | --- |
| National Key Research and Development Program of China | 2021YFD1801102 | Dengguo Wei |
| Fundamental Research Funds for the Central Universities | 2662023PY005 | Dengguo Wei |
| National Natural Science Foundation of China | 32473010 | Dengguo Wei |
| China Postdoctoral Science Foundation | 2024M761080 | Dehua Luo |
| Natural Science Foundation of Hubei Province | 2025AFB288 | Dehua Luo |
| HZAU-AGIS Cooperation Fund | SZYJY2022016 | Dengguo Wei |
| National Natural Science Foundation of China | 22077043 | Dengguo Wei |
| Natural Science Foundation of Hubei Province | 2021CFA061 | Dengguo Wei |
| State Key Laboratory of Agricultural Microbiology | AML2023B05 | Dengguo Wei |

The funders had no role in study design, data collection and interpretation, or the decision to submit the work for publication.

### Author contributions

Dehua Luo, Conceptualization, Formal analysis, Validation, Investigation, Writing – original draft, Writing – review and editing; Yingge Zheng, Investigation; Zhiyuan Huang, Zi Wen, Yingxiang Deng, Formal analysis; Lijun Guo, Qingling Li, Yuqing Bai, Visualization; Shozeb Haider, Validation, Writing – review and editing; Dengguo Wei, Conceptualization, Supervision, Validation, Writing – review and editing

## Author ORCIDs
Dehua Luo https://orcid.org/0009-0002-7827-7686
Shozeb Haider https://orcid.org/0000-0003-2650-2925
Dengguo Wei https://orcid.org/0000-0001-9923-789X

Reviewer #1 (Public review): https://doi.org/10.7554/eLife.103923.3.sa1
Reviewer #2 (Public review): https://doi.org/10.7554/eLife.103923.3.sa2
Reviewer #3 (Public review): https://doi.org/10.7554/eLife.103923.3.sa3
Author response https://doi.org/10.7554/eLife.103923.3.sa4

# Additional files

### Supplementary files
MDAR checklist

Supplementary file 1. Oligos used in this study.

Supplementary file 2. Characteristics of regions with different SHAPE reactivity and Shannon entropy.

Supplementary file 3. Location of regions with different SHAPE reactivity and Shannon entropy.

Supplementary file 4. Binding affinities of Compounds with 5'UTR-SL5 in the PEDV genome.

Supplementary file 5. PQSs with high conservation in the PEDV genome.

Supplementary file 6. Genomic locations and structural features of PQS-long chain regions.

Supplementary file 7. Sequence and characterization of target regions of siRNAs.

Supplementary file 8. Structural features of anti-SARS-CoV-2 siRNA target regions.

### Data availability
SHAPE-MaP datasets have been deposited in the NCBI Gene Expression Omnibus (GEO) under accession number GSE271098. Processed structural data (including the following files) are available at Zenodo (https://zenodo.org/doi/10.5281/zenodo.13743827): XML files with normalized SHAPE reactivities (SHAPE_react_rep1/2.xml); WIG files with Shannon entropies (GSE271098_PEDV_SHAPE_ Shannon_incell_rep1/2.wig.gz); CT file with the full genome secondary structure of PEDV (PEDV_ incell_secondary structure.ct).

The following datasets were generated:

| Author(s) | Year | Dataset title | Dataset URL | Database and Identifier |
|---|---|---|---|---|
| Luo D | 2024 | Exploiting functional regions in the viral RNA genome as druggable entities | https://doi.org/ 10.5281/zenodo. 13743827 | Zenodo, 10.5281/ zenodo.13743827 |
| Luo D | 2024 | Exploiting functional regions in the viral RNA genome as druggable entities | https://www.ncbi. nlm.nih.gov/geo/ query/acc.cgi?acc= GSE271098 | NCBI Gene Expression Omnibus, GSE271098 |

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
