## [Editor Report · eLife Assessment]

The study presents a **fundamental** advance in antiviral RNA research by adapting SHAPE-Map to chart the secondary structure of the porcine epidemic diarrhea virus (PEDV) genome in infected cells and pinpointing structurally conserved, accessible RNA elements as therapeutic targets. A broad, well-documented integration of biochemical probing, computational analysis, and functional validation provides **convincing** evidence that these regions are both biologically relevant and druggable. Beyond PEDV, the work offers a generalizable framework for RNA-guided antiviral discovery that will interest researchers in RNA therapeutics and viral genome biology.

---

## [Referee Report · Reviewer #1 (Public review)]

Summary:

This is a significant study because it adapts current methods to develop an approach for identifying promising targets for therapeutics in viral genomic RNA. The authors provide a wide array of data from different methods to help support their findings.

Strengths:

There are a number of strengths to highlight in this manuscript.

(1) The study uses a sophisticated technique (SHAPE-MaP) to analyze the PEDV RNA genome in situ, providing valuable insights into its structural features.

(2) The authors provide a strong rationale for targeting specific RNA structures for antiviral development.

(3) The study includes a range of experiments, including structural analysis, compound screening, siRNA design, and viral proliferation assays, to support their conclusions.

(4) Finally, the findings have potential implications for the development of new antiviral therapies against PEDV and other RNA viruses.

Overall, this interesting study highlights the importance of considering RNA structure when designing antiviral therapies and provides a compelling strategy for identifying promising RNA targets in viral genomes.

---

## [Referee Report · Reviewer #2 (Public review)]

Summary:

Luo et. al. use SHAPE-MaP to find suitable RNA targets in Porcine Epidemic Diarrhoea Virus. Results show that dynamic and transient structures are good targets for small molecules, and that exposed strand regions are adequate targets for siRNA. This work is important to segment the RNA targeting.

Strengths:

This work is well done and the data supports its findings and conclusions. When possible, more than one technique was used to confirm some of the findings.

Weaknesses:

The study uses a cell line that is not porcine (not the natural target of the virus). That being said, authors used a widely used cell line that has been used in similar studies.

---

## [Referee Report · Reviewer #3 (Public review)]

Summary:

This manuscript by Luo et al. applied SHAPE-Map to analyze the secondary structure of the Porcine Epidemic Diarrhoea Virus (PEDV) RNA genome in infected cells. By combining SHAPE reactivity and Shannon entropy, the study indicated that the folding of the PEDV genomic RNA was nonuniform, with the 5' and 3' untranslated regions being more compactly structured, which revealed potentially antiviral targetable RNA regions. Interestingly, the study also suggested that compounds bound to well-folded RNA structures in vitro did not necessarily exhibit antiviral activity in cells, because the binding of these compounds did not necessarily alter the functions of the well-folded RNA regions. Later in the manuscript, the authors focus on guanine-rich regions, which may form G-quadruplexes and be potential targets for small interfering RNA (siRNA). The manuscript shows the binding effect of Braco-19 (a G-quadruplex-binding ligand) to a predicted G4 region in vitro, along with the inhibition of PEDV proliferation in cells. This suggests that targeting high SHAPE-high Shannon G4 regions could be a promising approach against RNA viruses. Lastly, the manuscript identifies 73 single-stranded regions with high SHAPE and low Shannon entropy, which demonstrated high success in antiviral siRNA targeting.

Strengths:

The paper presents valuable data for the community. Additionally, the experimental design and data analysis are well documented.

Weaknesses:

I have no further comments after the authors validated their concept by adding the ThT fluorescence assay in the revised version.

---

## [Author Response]

The following is the authors’ response to the previous reviews

**Public Reviews:**

**Reviewer #1 (Public review):**
Summary:This study investigates the potential of targeting specific regions within the RNA genome of the Porcine Epidemic Diarrhea Virus (PEDV) for antiviral drug development. The authors used SHAPE-MaP to analyze the structure of the PEDV RNA genome in infected cells. They categorized different regions of the genome based on their structural characteristics, focusing on those that might be good targets for drugs or small interfering RNAs (siRNAs).They found that dynamic single-stranded regions can be stabilized by compounds (e.g., to form G-quadruplexes), which inhibit viral proliferation. They demonstrated this by targeting a specific G4-forming sequence with a compound called Braco-19. The authors also describe stable (structured) single-stranded regions that they used to design siRNAs showing that they effectively inhibited viral replication.Strengths:There are a number of strengths to highlight in this manuscript.(1) The study uses a sophisticated technique (SHAPE-MaP) to analyze the PEDV RNA genome in situ, providing valuable insights into its structural features.(2) The authors provide a strong rationale for targeting specific RNA structures for antiviral development.(3) The study includes a range of experiments, including structural analysis, compound screening, siRNA design, and viral proliferation assays, to support their conclusions.(4) Finally, the findings have potential implications for the development of new antiviral therapies against PEDV and other RNA viruses.Overall, this interesting study highlights the importance of considering RNA structure when designing antiviral therapies and provides a compelling strategy for identifying promising RNA targets in viral genomes.Weaknesses:I have some concerns about the utility of the 3D analyses, the effects of their synonymous mutants on expression/proliferation, a potentially missed control for studies of mutants, and the therapeutic utility of the compound they tested vs. Gquadruplexes.

We thank the reviewer for their positive assessment and insightful comments. Below, we address each point of concern:

(1) The utility of the 3D analyses:

In the revised manuscript, we have toned down this discussion and moved Figure 3A to the supplementary materials to reduce any sense of fragmentation in the overall story. While SHAPE-MaP technology is mature and convenient to use and can indeed capture some RNA structural elements with special functions in certain case; we acknowledge that its application for 3D analyses requires further validation. We believe this approach will become more prevalent in future research.

(2) The effects of synonymous mutants on expression/proliferation:

In the PEDV genome, the PQS1 mutation site encodes lysine (AAG). Given that lysine has only two codons (AAG and AAA), the G3109A synonymous mutation represented our sole viable option. Published studies (Ding et al., 2024) confirm that neither AAG nor AAA are classified as rare or dominant codons in mammalian cells. Therefore, the observed changes in viral proliferation levels are likely to stem from alterations in RNA secondary structure rather than codon usage effects.

REFERENCES:

Ding W, Yu W, Chen Y, et al. Rare codon recoding for efficient noncanonical amino acid incorporation in mammalian cells. Science. 2024;384(6700):1134-1142.

(3) Potentially missed control for studies of mutants:

In the revised manuscript, we have incorporated additional control experiments evaluating Braco-19's therapeutic effects on the PQS3 mutant strain (Figure 4 – figure supplement 3)：

(4) The therapeutic utility of Braco-19 vs. G-quadruplexes:

While Braco-19 is indeed a broad-spectrum G4 ligand, our data clearly show that not all PQSs in the viral genome can form G4 structures. Our findings primarily provide proof-of-concept that sequences with high G4-forming potential in viral genomes represent viable targets for antiviral therapy. Future studies could leverage SHAPEguided structural insights to design ligands with enhanced specificity for viral G4s, potentially improving therapeutic utility while minimizing off-target effects.

**Reviewer #2 (Public review):**
Summary:Luo et. al. use SHAPE-MaP to find suitable RNA targets in Porcine Epidemic Diarrhoea Virus. Results show that dynamic and transient structures are good targets for small molecules, and that exposed strand regions are adequate targets for siRNA. This work is important to segment the RNA targeting.Strengths:This work is well done and the data supports its findings and conclusions. When possible, more than one technique was used to confirm some of the findings.Weaknesses:The study uses a cell line that is not porcine (not the natural target of the virus).

We thank the reviewer for their insightful comments and recognition of our study's value. The most commonly employed cell models for in vitro PEDV studies are monkey-derived Vero E6 cells and porcine PK1 cells. However, PEDV (particularly our strain) exhibits significantly lower replication efficiency in PK1 cells compared to Vero cells, and no cytopathic effects were observed in PK1 cells. In our preliminary attempts to perform SHAPE-MaP experiments using infected PK1 cells, the sequencing data showed less than 0.03% alignment to the PEDV genome, rendering subsequent analysis and downstream experiments unfeasible.

**Reviewer #3 (Public review):**
Summary:This manuscript by Luo et al. applied SHAPE-Map to analyze the secondary structure of the Porcine Epidemic Diarrhoea Virus (PEDV) RNA genome in infected cells. By combining SHAPE reactivity and Shannon entropy, the study indicated that the folding of the PEDV genomic RNA was nonuniform, with the 5' and 3' untranslated regions being more compactly structured, which revealed potentially antiviral targetable RNA regions. Interestingly, the study also suggested that compounds bound to well-folded RNA structures in vitro did not necessarily exhibit antiviral activity in cells, because the binding of these compounds did not necessarily alter the functions of the well-folded RNA regions. Later in the manuscript, the authors focus on guanine-rich regions, which may form G-quadruplexes and be potential targets for small interfering RNA (siRNA). The manuscript shows the binding effect of Braco-19 (a G-quadruplex-binding ligand) to a predicted G4 region in vitro, along with the inhibition of PEDV proliferation in cells. This suggests that targeting high SHAPE-high Shannon G4 regions could be a promising approach against RNA viruses. Lastly, the manuscript identifies 73 singlestranded regions with high SHAPE and low Shannon entropy, which demonstrated high success in antiviral siRNA targeting.Strengths:The paper presents valuable data for the community. Additionally, the experimental design and data analysis are well documented.Weakness:The manuscript presents the effect of Braco-19 on PQS1, a single G4 region with high SHAPE and high Shannon entropy, to suggest that "the compound can selectively target the PQS1 of the high SHAPE-high Shannon region in cells" (lines 625-626). While the effect of Braco-19 on PQS1 is supported by strong evidence in the manuscript, the conclusion regarding the G4 region with high SHAPE and high Shannon entropy is based on a single target, PQS1.

We thank the reviewer for their positive assessment of our methodology and dataset. We propose that dynamic RNA structures in high SHAPE-high Shannon regions, when stabilized by small molecules, can serve as viable targets for antiviral therapy. Gquadruplexes represent a characteristic type of such dynamic structures that compete with local stem-loop formations in the genome. While we identified seven highly conserved PQSs in the PEDV genome, only PQS1 was located within a high SHAPEhigh Shannon region. To further validate this concept, we have supplemented the revised manuscript with Thioflavin T (ThT) fluorescence turn-on assays (Figures 3D, 3E, and Figure 3 – figure supplement 6), which provide additional evidence for the differential G4-forming capabilities of PQSs across regions with distinct structural features.

**Recommendations for the authors:**

**Reviewer #1 (Recommendations for the authors):**
Major Comments:(1) It could be valuable for the authors to spend some more effort comparing their approach to siRNA target discovery and design to current methods for siRNA design. It would be good to highlight which components are novel, and which might offer superior performance with respect to other existing methods.

We thank the reviewer for highlighting this important point. In response, we have rewritten the relevant section in the discussion:

“Our approach uniquely integrates in situ RNA structural data (SHAPE reactivity and Shannon entropy) to prioritize siRNA targets within stable single-stranded regions (high SHAPE reactivity, low Shannon entropy), which are experimentally validated as accessible in infected cells. This represents a significant departure from traditional siRNA design methods that rely primarily on sequence conservation, thermodynamic rules (e.g., Tuschl rules), or in vitro structural predictions (Ali Zaidi et al., 2023; Qureshi et al., 2018; Tang and Khvorova, 2024),which may not accurately reflect intracellular RNA accessibility. Bowden-Reid et al. designed 39 antiviral siRNAs against various SARS-CoV-2 variants based on sequence conservation, ultimately identifying 8 highly effective sequences (Bowden-Reid et al., 2023). Notably, five of these effective sequences targeted regions that were located in high SHAPE-high Shannon regions according to SARS-CoV-2 SHAPE datasets (Supplementary Table 8) (Manfredonia et al., 2020). This independent finding aligns perfectly with our conclusions and demonstrates that SHAPE-based siRNA design outperforms sequence/structureagnostic approaches, at least in terms of significantly improving antiviral siRNA screening efficiency. Given the growing availability of SHAPE datasets for numerous viruses, we are confident that our methodology will facilitate more precise design of antiviral siRNAs.”

(2) The section targeting their discovered G4 structure with Braco-19 is interesting, particularly showing effects on viral proliferation; however, it's not clear to me how this compound could be used therapeutically against PEDV, as it is a non-selective binder of G4 structures. Their results are good support for the presence and functionality of a G4 structure in PEDV, but I don't see any strategy outlined in the manuscript on how this could be specifically targeted with Braco-19.

While Braco-19 is indeed a broad-spectrum G4 ligand, our data demonstrate that not all PQSs in the viral genome can form G4 structures under physiological conditions. Our results specifically show that Braco-19 exerts its anti-PEDV activity by targeting PQS1, which is located in a high SHAPE-high Shannon entropy region. This target specificity was further confirmed by the complete resistance of the PQS1mut strain (lacking G4-forming ability) to Braco-19 treatment in our in vitro assays.

Additionally, previous studies have reported that during rapid viral replication, viral RNA accumulates to levels that significantly exceed host RNA concentrations. This "concentration advantage" suggests that G4 ligands like Braco-19 would preferentially bind viral G4 structures over host targets, thereby enhancing their antiviral specificity in vivo. In summary, our data provide proof-of-concept that viral genomic regions with high G4-forming potential - particularly those in high SHAPE-high Shannon entropy regions - represent promising targets for antiviral therapy.

(3) The section where they proposed 3D RNA structures based on sequence similarity feels "tacked on" and I don't see how it adds to the overall story. The authors identify a short RNA hairpin in the PEDV genome with some sequence similarity to the CPEB3 nuclease P4 hairpin. However, they don't provide any evidence that this motif functions in a similar way or that it's important for the virus's life cycle. They also don't explain how this similarity could be exploited for antiviral drug development. It's not clear whether targeting this motif would have any effect on the virus. It's interesting that these two sequences share nucleotides, but it's unlikely that they share any homology...perhaps they convergently evolved (or were captured), but the similarity could also be coincidental.

We appreciate the reviewer's insightful observation regarding this section. While our intention was to demonstrate that flexible conformations in high SHAPE-high Shannon regions could potentially be targeted, we acknowledge that extensive discussion of these motifs' functions would exceed the scope of this study, resulting in some disconnection from the main narrative. In response to this valuable feedback, we have consequentially removed it from the manuscript.

(4) The authors should consider the optimality of the synonymous mutation (G3109A) that they introduced, as G3109A could swap a rare codon for a more optimal one. Even though the protein sequence is unaffected, the translation rate (and ability to proliferate) could be very different due to altered codon optimality. Additionally, to show the inactivity of the PQS3 mutant, the Braco-19 treatment studies performed on the PQS1 mutants could be repeated with PQS3 - using this as a control for these experiments.

We appreciate the reviewer's insightful comment regarding codon optimization. In the PEDV genome, the PQS1 mutation site encodes lysine (AAG). Since lysine has only two codons (AAG and AAA), the G3109A synonymous mutation was our only viable option. Published literature (Ding et al. 2024) confirms that neither AAG nor AAA are classified as either preferred or rare codons in mammalian cells. Therefore, this substitution should have minimal direct impact on translation efficiency. Compared to nonsynonymous mutations that would alter amino acid sequences, we believe this synonymous mutation represents the optimal approach for maintaining native protein function while introducing the desired structural modification.

REFERENCES:

Ding W, Yu W, Chen Y, et al. Rare codon recoding for efficient noncanonical amino acid incorporation in mammalian cells. Science. 2024;384(6700):1134-1142.

In the revised version, we have added control experiments showing the inhibitory activity of Braco-19 against the PQS3 mutant strain (Figure 4—figure supplement 3C) and discussed it in the results section.

“Furthermore, as a control, we observed nearly identical inhibitory activity of Braco19 against both the PQS3 mutant strain (AJ1102-PQS3mut) and wild-type virus (Figure 4—figure supplement 3C), demonstrating the specificity of Braco-19's action on PQS1.”

Minor Comments:(5) The authors' description of the Shannon Entropy could be improved. The current description makes it seem like the Shannon Entropy only provides information on base pairing, however, the Shannon entropy quantifies the uncertainty of structural states at each position and is calculated based on the probabilities of the different states (paired or unpaired) that a nucleotide can adopt.

We have revised the description of Shannon entropy in the manuscript：

"The pairing probability of each nucleotide derived from SHAPE reactivities was subsequently used to calculate Shannon entropy. Regions with high Shannon entropy may adopt alternative conformations, while those with low Shannon entropy correspond to either well-defined RNA structures or persistently single-stranded regions (MATHEWS, 2004; Siegfried et al., 2014)."

(6) The overall writing of the manuscript is very good, but there are some minor grammatical issues throughout, e.g., here are some of the ones that I caught:a) Lines 71-3: "various types of RNA structures such as hairpin structure, RNA singlestrand, RNA pseudoknot and RNA G-quadruplex (G4)" - the examples should be plural and, rather than "hairpins" (or in addition), perhaps add "helixes" to be more generically correct(?).

We have revised the relevant description:

"various types of RNA structures such as stem-loop structures (with double-helical stems), RNA single-strand, RNA pseudoknot and RNA G-quadruplex (G4)"

b) Lines 74-5: "Of these, RNA G4 has shown considerable promise because of the high stability and modulation by small molecules" should be "Of these, RNA G4 has shown considerable promise because of its high stability and ability for modulation by small molecules."

We have revised the sentence:

“Of these, RNA G4 has shown considerable promise because of its high stability and ability for modulation by small molecules.”

c) Line 76: "have" should be "has".

We have revised the sentence.

d) Lines 104-5 (and elsewhere): "frameshift stimulation element (FSE)" should be "frameshift stimulatory element (FSE)".

We have revised the sentence.

e) Lines 428-9: following the Manfredonia's methods" should be "following Manfredonia's method" or "following the Manfredonia method".

We have made the appropriate edit.

These edits ensure grammatical accuracy and consistency with standard scientific terminology. We appreciate the reviewer's attention to detail, which has significantly improved the clarity of our manuscript.

**Reviewer #2 (Recommendations for the authors):**
(1) There are some important references missing, on shape-seq from Julius Lucks.

We have added citations to the foundational work by Lucks et al. (2011, PNAS) that pioneered in vitro RNA structure probing using SHAPE-seq.

(2) Describe the acronym "SHAPE",

We have now included the full name of SHAPE:“Selective 2’-Hydroxyl Acylation and Primer Extension”.

(3) Line 81: 2"-hydroxyl-selective - the prime is incorrect.

We thank the reviewer for catching this technical error. We have corrected "2"hydroxyl" to "2'-hydroxyl".

(4) Explaining a bit better how shape reagent works would be beneficial (one sentence should suffice).

We have revised the Introduction section:

“SHAPE reagents like NAI selectively modify flexible, unpaired 2′-OH groups in RNA, and these modifications are detected as mutations during reverse transcription, enabling precise mapping of RNA secondary structures through sequencing.”

(5) Line 128: cite the paper that introduced NAI.

We have now properly cited the original publication introducing NAI（Spitale et al., 2012）.

(6) Line 243: Can you describe what the compound is?

The compound is Braco-19. This has now been included in the methods section.

(7) Line 272: describe what 3Dpol is and the source of it.

We have supplemented the relevant information as follows:

"3Dpol (recombinant RNA-dependent RNA polymerase; Abcam, ab277617, 0.02 mg/reaction)"

(8) Figure 1 legend: For both C and D, the explanation of the G4 structure and the RISC complex should be added, otherwise, it becomes unclear why they are there.

We have revised the captions for Figure 1 as follows:

"(A) Well-folded regions (low SHAPE reactivity and low Shannon entropy; 26.40% of genome). These regions represent stably folded RNA structures with minimal conformational flexibility, likely serving as structural scaffolds or functional elements in viral replication. (B) Dynamic structured regions (low SHAPE reactivity and high Shannon entropy; 11.70% of genome). These conformationally plastic domains likely mediate regulatory switches between alternative secondary structures during infection. (C) Dynamic unpaired regions (high SHAPE reactivity and high Shannon entropy; 26.90% of genome). These regions are prone to form non-canonical nucleic acid structures (e.g., G-quadruplexes), which can be stabilized by small-molecule ligands to inhibit viral replication. (D) Persistent unpaired regions (high SHAPE reactivity and low Shannon entropy; 9.67% of genome). These regions are more accessible for siRNA binding, facilitating recruitment of Argonaute proteins and Dicer to form the RNAinduced silencing complex (RISC) for targeted cleavage."

(9) Figure S2 panel A should be in Figure 1. This is a nice picture showing the backbone of the research.

In the revised manuscript, we have reorganized Figure 1 and Figure S2 by incorporating the SHAPE-MaP workflow diagram (previously Figure S2A) into Figure 1 as panel (A):

(10) Please add the citation to Braco-19.

We have now added the appropriate citation for Braco-19 (Gowan et al., 2002) in the revised manuscript.

(11) Figure 5 legend: could you add in parenthesis the what ds means (and call Figure S28).

We appreciate the reviewer's attention to detail. In the revised manuscript, we have clarified the abbreviations in the Figure 5 legend: ss (single-stranded targeting siRNAs); ds (dual-stranded targeting siRNAs).

(12) Line 107: I would argue that the "stabilization of a G4" inhibited viral proliferation. And that supports the point of the paper, that a small molecule that stabilizes the G4 can be used to reduce viral replication. I suggest emphasizing this thorough the paper.

We fully concur with the reviewer's insightful perspective. In the revised manuscript, we have comprehensively strengthened the point of 'G4 stabilization' as an antiviral mechanism through the following enhancements:

(1) In the Results section: We present Thioflavin T (ThT) fluorescence assays demonstrating the G4-forming capability of PQSs in the full-length PEDV genomic RNA context:

“These findings indicate that although most PQSs can form G4 structures in vitro, PQS1—located in the high SHAPE-high Shannon entropy region—demonstrates the most robust G4-forming capability when competing with local secondary structures in the genomic context.”

(2) In the Results section: The inclusion of Braco-19 inhibition assays using PQS3 mutant virus as control provides robust evidence that Braco-19 exerts its antiviral effects specifically through PQS1 stabilization:

“Furthermore, as a control, we observed nearly identical inhibitory activity of Braco-19 against both the PQS3 mutant strain (AJ1102-PQS3mut) and wild-type virus, demonstrating the specificity of Braco-19's action on PQS1.”

(3) In the Discussion section: We have rewritten the mechanistic interpretation to emphasize:

"Crucially, Braco-19 showed no inhibitory activity against the PQS1-mutant strain while maintaining potent activity against the PQS3-mutant strain (Figure 4E, Figure 4—figure supplement 3C). This suggests that the compound can selectively target the PQS1 of the high SHAPE-high Shannon region in cells."

(13) For PQS1, it's suggested that it is indeed a competing and transient conformation that forms the G4. I wonder if using an extended PQS1 (perhaps what is shown in Figure 3E) and using fluorescence, and/or K+ vs Li+, and/or in-vitro SHAPE could tell us more about this dynamic structure. Thioflavin T or any other fluorescent molecule that binds to G4s could be easily used to show how the formation of G4 may happen or not. In addition, how Braco-19 could really lock the dynamic structure in-vitro as well. I think the field would benefit from a deeper investigation of it.

To address the dynamic competition between G4 and alternative RNA conformations, we performed Thioflavin T (ThT) fluorescence turn-on assay (now in Figure 3D-E and Figure 3—figure supplement 6) under physiological K^+^ conditions (100 mM), with PRRSV-G4 RNA as a positive control. This reads as:

“To validate whether SHAPE analysis could reflect the competitive conformational folding of PQSs in the PEDV genome, we performed in vitro transcription to obtain local intact structures containing PQSs within dynamic single-stranded regions and stable double-stranded regions (Table S6). Thioflavin T (ThT) fluorescence turn-on assays were conducted under physiological K^+^ conditions (100 mM), with the G4 sequence of porcine reproductive and respiratory syndrome virus (PRRSV) serving as a positive control (Control-G4)(Fang et al., 2023). The results demonstrated that for short PQSs sequences containing only G4-forming motifs (Table S7), PQS1, PQS3, PQS4, and PQS6 all induced significant ThT fluorescence enhancement (Figure 3D-E, Figure 3—figure supplement 6), confirming their ability to form G4 structures. However, in long RNA fragments encompassing PQSs and their flanking sequences, only PQS1 and PQS4 exhibited pronounced ThT fluorescence responses (Figure 3DE), whereas PQS2, PQS3, and PQS6 showed negligible signals (Figure 3E, Figure 3— figure supplement 6). Notably, the PQS1-long chain displayed the strongest fluorescence signal, while its mutant counterpart (PQS1mut-long chain) exhibited the lowest background fluorescence (Figure 3D). These findings indicate that although most PQSs can form G4 structures in vitro, PQS1—located in the high SHAPE-high Shannon entropy region—demonstrates the most robust G4-forming capability when competing with local secondary structures in the genomic context. Therefore, PQS1 was selected for further structural and functional validation.”

(14) Figure S29 is nice and informative. Consider moving it to the main text.

We appreciate the reviewer's positive assessment of Figure S29. Now we have renamed this figure as "Figure 5—Supplement 2".